# LDT: Layer-Decomposition Training Makes Networks More Generalizable

**Zai-Zuo Tang, Zong-Qi Yang, Yu-Bin Yang** *
State Key Laboratory of Novel Software Technology
Nanjing University
Nanjing, China
{tangzz, zongqiyang}@smail.nju.edu.cn, yangyubin@nju.edu.cn

## Abstract

Domain generalization methods can effectively enhance network performance on test samples with unknown distributions by isolating gradients between unstable and stable parameters. However, existing methods employ relatively coarse-grained partitioning of stable versus unstable parameters, leading to misclassified unstable parameters that degrade network feature processing capabilities. We first provide a theoretical analysis of gradient perturbations caused by unstable parameters. Based on this foundation, we propose Layer-Decomposition Training (LDT), which conducts fine-grained layer-wise partitioning guided by parameter instability levels, substantially improving parameter update stability. Furthermore, to address gradient amplitude disparities within stable layers and unstable layers respectively, we introduce a Dynamic Parameter Update (DPU) strategy that adaptively determines layer-specific update coefficients according to gradient variations, optimizing feature learning efficiency. Extensive experiments across diverse tasks (super-resolution, classification, semantic segmentation) and architectures (Transformer, Mamba, CNN) demonstrate LDT's superior generalization capability. Our code is available at https://github.com/ZaizuoTang/LDT.

## 1 Introduction

With advances in neural networks and computing hardware, neural network-based methods have become dominant across visual tasks, spanning both high-level vision (image classification(Lee et al., 2025), semantic segmentation(Zhang et al., 2025), object detection(Chen et al., 2025), point cloud segmentation(TangZaizuo et al., 2023)) and low-level vision (image super-resolution(Zhou et al., 2023), image generation(Shi et al., 2024)). However, their superior performance critically depends on the assumption that training and test data share similar distributions.

In real-world scenarios, due to variations in illumination conditions and imaging devices, the test sample distribution (target domain) and training sample distribution (source domain) exhibit significant differences, termed as domain shift. Domain shift causes networks that perform well on the source domain to suffer severe performance degradation on the target domain, which greatly limits their application in high-confidence-demand tasks (e.g., medical diagnosis, autonomous driving, etc.).

Consequently, domain generalization methods have emerged (Kumar et al., 2022; Pahk et al., 2025; Wang et al., 2024b), which effectively enhance the generalization capability of networks through data augmentation strategies (Vaish et al., 2024b; Xu et al., 2025; Zheng et al., 2024) and explicit learning of domain-invariant features (Huang et al., 2024; Li et al., 2024b). Data augmentation-based domain generalization methods fall into two categories: (1) Input sample augmentation improves network robustness to different degradation patterns through rotations, crops, and frequency-domain perturbations of input samples. (2) Network architecture augmentation improves parameter generalization via perturbations on specific layers or channels (e.g., Dropout (Hinton et al., 2012)). Explicit domain-invariant feature learning methods enhance model robustness by decomposing input

---
*Corresponding author

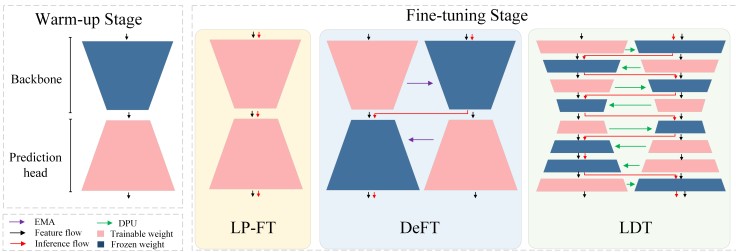

Figure 1: Comparison with existing methods. Existing methods consist of two stages: a warm-up stage for initializing the prediction head, and a fine-tuning stage. During fine-tuning, LP-FT (Kumar et al., 2022) fine-tunes the entire network, while DeFT (Pahk et al., 2025) treats the backbone network as stable layers and the prediction head as unstable layers, constructing primary and auxiliary networks with cross-freezing of backbone and prediction head components to stabilize gradient updates. Our proposed LDT method achieves finer-grained hierarchical separation of stable and unstable layers and incorporates the dynamic parameter update (DPU) strategy into the parameter update process. Notably, for low-level vision tasks such as super-resolution (SR), the network's prediction head is replaced with an upsampling module.

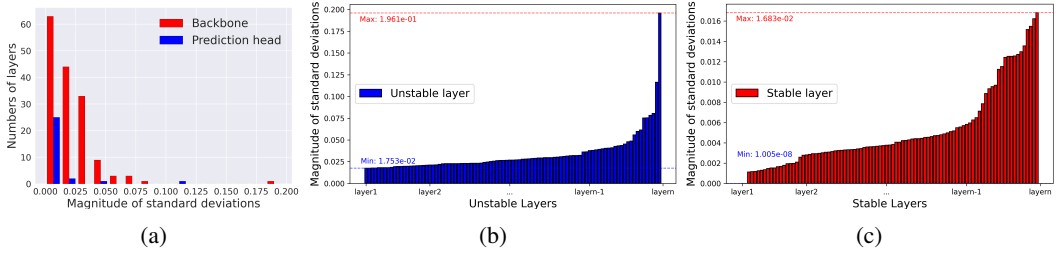

Figure 2: Gradient stability analysis. (a) Gradient stability analysis for each layer in both the backbone network and prediction head. We feed 600 samples through the network to collect layer-wise gradients without performing parameter updates, then compute the standard deviation of each layer's gradients across these samples. (b) Gradient stability analysis for unstable layers. (c) Gradient stability analysis for stable layers. We statistically analyze gradient variations in stable layers and unstable layers partitioned by LDT, respectively.

features into domain-invariant and domain-specific components, then selectively emphasis domain-invariant representations while suppressing domain-specific variations. (For related work on domain generalization methods, refer to Appendix A.)

Although existing domain generalization methods have extensively investigated input samples, network architectures, and intermediate features, their exploration of parameter correlations in domain generalization tasks remains limited.

Two recent works, LPFT (Kumar et al., 2022) and DeFT(Pahk et al., 2025), have conducted preliminary exploration of parameter-to-parameter correlations in networks, focusing on network generalization in fine-tuning scenarios. There exists a randomly initialized prediction head and a backbone network pretrained with large-scale data (ImageNet (Deng et al., 2009)), where the backbone possesses strong feature processing capability and generalization performance. They argue that during fine-tuning, the random parameter distribution in the uninitialized prediction head will perturb parameter updates in the backbone network, ultimately compromising the network's overall performance and generalization capability. As shown in Figure.1, both LPFT and DeFT methods divide the training process into a warm-up stage and a fine-tuning stage. During the warm-up stage, they freeze the backbone network to prevent interference from the prediction head. In the fine-tuning stage, the LPFT method fine-tunes all network parameters to maximize feature learning efficiency. The DeFT method maintains isolation between the prediction head and backbone network during fine-tuning by constructing a dual-branch architecture with cross-freezing of the backbone and prediction head, thereby preventing interference from the prediction head.

Gradients represent the change in network parameters in response to current input samples. The network contains unstable parameters that exhibit extreme sensitivity to input feature distributions - minor variations can trigger severe fluctuations in these parameters. These unstable parameters constitute the fundamental factor impairing network generalization performance. Through theoretical

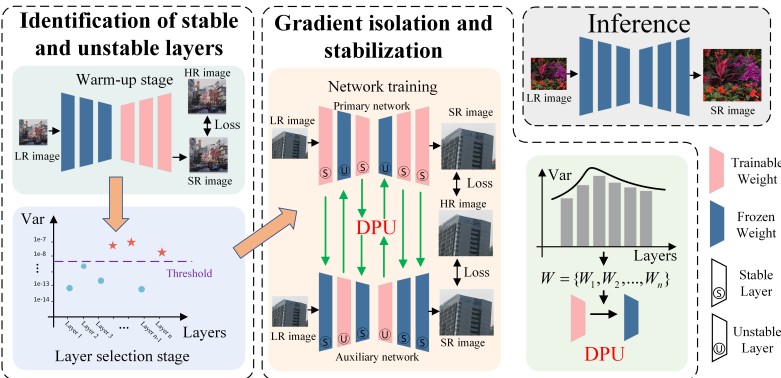

Figure 3: Overall framework.

analysis (Appendix B), we demonstrate these unstable parameters significantly influence parameter updates throughout all network layers by interfering with gradient propagation.

Due to the stochastic nature of unstable parameter fluctuations, their gradient directions are typically more random, exhibiting higher variance. Therefore, as shown in Figure.2, we statistically analyze gradient variations of network parameters at the layer-wise level across different input samples, revealing the following two issues:

- Insufficient granularity in partitioning stable and unstable parameters: existing methods such as LP-FT (Kumar et al., 2022) and DeFT (Pahk et al., 2025) designate the backbone network as the stable layers and the prediction head as the unstable layers. However, as shown in Figure 2a, certain layers within the backbone network demonstrate higher gradient variance compared to the prediction head, indicating that some backbone layers are actually less stable than the head. These unstable parameters in the backbone substantially influence parameter updates across all other layers. As a result, existing partitioning methods based on the backbone-head distinction operate at an insufficient granularity, leading to misclassification issues.

- Inadequate adaptability in parameter updates: even within the unstable layers, significant disparities in gradient variance exist (with the ratio between the minimum and maximum standard deviations approaching 10×, as shown in Figure 2b). Existing methods apply the same update strategies to all layers, despite their varying gradient standard deviations, which inevitably leads to underutilization of information. This issue is even more pronounced among stable layers, as illustrated in Figure 2c.

To address the low partitioning granularity issue, we propose the Layer-Decomposition Training (LDT) strategy. LDT performs layer-wise partition of stable and unstable layers based on gradient variance of parameters across network layers, and employs subsequent cross-freezing to prevent gradient interference from unstable layers to stable ones, effectively enhancing network generalization. For the parameter update adaptability problem, we further introduce the Dynamic Parameter Update (DPU) strategy within LDT framework. DPU projects gradient variance into parameter update coefficients, enabling networks to self-adaptively determine update ranges, thereby further improving generalization performance.

Our main contributions can be summarized as follows:

- We first provide a theoretical analysis of perturbation effects from unstable parameters to stable parameters. Building on this foundation, we propose Layer-Decomposition Training (LDT), which mitigates perturbations from unstable layers during training through explicit separation of stable and unstable layers, effectively enhancing domain generalization.

- We develop a Dynamic Parameter Update (DPU) strategy that adapts update coefficients based on fluctuation amplitudes, demonstrating superior adaptability compared to conventional EMA methods.

Table 1: Symbol Definitions

| Symbol | Meaning/Description | Symbol | Meaning/Description |
|--------|--------------------|--------|--------------------|
| $PM$ | Primary Network | $AM$ | Auxiliary Network |
| $PL$ | Primary Network Layer | $AL$ | Auxiliary Network Layer |
| $P\theta$ | Parameters of the Primary Network | $A\theta$ | Parameters of the Auxiliary Network |
| $\tilde{\cdot}$ | Frozen / Non-trainable | $\Delta$ | Gradient |
| $W$ | Parameter Update Coefficient | $x$ | Input Features |
| $y$ | Prediction Result | $\overline{y}$ | Ground Truth Label |
| $\cdot^P$ | Primary | $\cdot^A$ | Auxiliary |
| $\cdot^S$ | Stable | $\cdot^U$ | Unstable |
| $\cdot^{So}$ | Source Domain | $\cdot^T$ | Target Domain |

- Our method is architecture-agnostic and task-agnostic, validated across diverse vision tasks (both high-level and low-level) and architectures (Transformer, Mamba, CNN), demonstrating LDT's general effectiveness.

## 2 METHOD

### 2.1 PROBLEM DEFINITION AND NOTATION

Given a source domain $D^{So} \in \{D_1^{So}, \cdots, D_{n^{So}}^{So}\}$ and a target domain $D^T \in \{D_1^T, \cdots, D_{n^T}^T\}$, which comprise $n^{So}$ and $n^T$ sub-datasets respectively and are disjoint $D^{So} \cap D^T = \emptyset$. We perform supervised training of network $M$ on the source domain $D^{So}$ to enhance its generalization performance, i.e., achieve satisfactory performance on target domain $D^T$ that are unknown during trainin

For clarity, Table 1 provides a unified definition of the notations used throughout this paper.

### 2.2 OVERALL FRAMEWORK

As shown in Figure.3, the training procedure of LDT consists of two components: (1) Identification of stable and unstable layers, and (2) Gradient isolation and stabilization. The identification component contains two sub-stages: the Warm-Up stage initializes the prediction head, while the Layer Selection stage performs fine-grained layer-wise partitioning of stable and unstable layers. In the gradient isolation component, the network is duplicated into two copies (primary and auxiliary networks) with alternating layer freezing: unstable layers are frozen in the primary network while stable layers remain trainable; conversely, stable layers are frozen in the auxiliary network while unstable layers are made trainable. During simultaneous training of both networks, the proposed Dynamic Parameter Update strategy (DPU, Section 2.4) adaptively adjusts the frozen parameters in both networks. After training completion, frozen layers from both networks are extracted and combined into a new composite network for test-time inference. (See Appendix E for the complete training pipeline pseudocode)

### 2.3 LAYER-DECOMPOSITION TRAINING (LDT)

#### 2.3.1 MOTIVATION

The gradient represents the change in network parameters induced by current inputs, where larger gradients indicate more significant parameter modifications. When a sample is fed into the network, large gradients primarily arise from two scenarios: (1) parameters possess strong feature processing capabilities and exhibit stronger feedback to current inputs; (2) parameters are overly sensitive to input samples, where minor input variations cause large parameter fluctuations, thereby generating large gradients. In the second scenario, the parameters exhibit high sensitivity to domain shift, severely degrading performance on target domains with unknown distributions. **More critically, during network training, unstable parameters (second scenario) interfere with the gradients of stable parameters (see Appendix B for theoretical proof).** Furthermore, through iterative training (forward propagation-gradient computation-parameter updates), this interference

becomes progressively amplified. Therefore, we aim to decouple unstable parameters' interference with stable ones while stabilizing updates of unstable parameters.

### 2.3.2 IDENTIFICATION OF UNSTABLE AND STABLE LAYERS

Since the training sample distribution is relatively uniform, (as discussed in the preceding subsection) the first case results in stable, directionally consistent parameter updates with low gradient variance. In contrast, parameters in the second scenario exhibit random fluctuations with stochastic update directions and high variance. Therefore, we propose to partition parameters based on variance, using layers as the partition unit - treating layers with high gradient variance as unstable and those with low variance as stable. Notably, to prevent interference from randomly initialized weights when obtaining gradient variance, a warm-up stage is introduced that initializes the network using a subset of the source domain.

The source domain samples are partitioned into two subsets $D^S = \{D^{S_1}, D^{S_2}\}$. During the warm-up stage, one source domain subset $D^{S_1}$ is used for network parameter initialization. After network initialization, the other source domain subset $D^{S_2}$ is fed into the network, where gradients at each layer are preserved without parameter updates. Subsequently, LDT calculates the variance of each layer's gradients across samples from source domain subset $D^{S_2}$, defining layers with high variance as unstable layers and those with low variance as stable layers,

$$Name^U = Top\_N(Var, Ratio^U, M), \tag{1}$$

$$Name^S = Name^{All} - Name^U, \tag{2}$$

where $Name^{All}$ is the set of all layer names in network $M$, $Top\_N$ selects the top-ranked layer names based on each layer's gradient variance to define unstable layers $Name^U$. The stable layer name set $Name^S$ is the complement of unstable layer $Name^U$ name set within the full set $Name^{All}$. $Ratio^U$ indicates the ratio of unstable layers.

### 2.3.3 GRADIENT ISOLATION AND STABILIZATION

As shown in Figure.3, to stabilize parameter updates in unstable layers, inspired by (Kumar et al., 2022; Pahk et al., 2025), the LDT method adopts a parallel dual-branch training strategy. It duplicates the initialized network from the warm-up stage into two copies: the primary network $PM$ and auxiliary network $AM$. Subsequently, LDT freezes the unstable layers in the primary network, preventing gradient updates via the loss function, while employing the Exponential Moving Average (EMA) algorithm to update primary network's frozen unstable layer parameters using those from auxiliary network's unfrozen unstable layers. This mechanism aims to stabilize the unstable layers in primary network by leveraging multi-timestep parameters from auxiliary network's unstable layers,

$$\tilde{P}\theta^U_{t+1} = W_f \times \tilde{P}\theta^U_t + (1 - W_f) \times A\theta^U, \tag{3}$$

where $\tilde{P}\theta^U$ denotes the parameters of the frozen unstable layers in the primary network, $A\theta^U$ corresponds to the parameters of the unfrozen unstable layers in the auxiliary network, and $W_f$ is the parameter update coefficient. A value of $W_f$ closer to 1 indicates that more timesteps of the auxiliary network's unstable layer parameters $A\theta^U$ are required to induce changes in the primary network's unstable layer parameters $\tilde{P}\theta^U$.

To eliminate gradient interference from unstable layers to stable layers, the stable layers in the auxiliary network are frozen, allowing loss function gradients to update only the unstable layers in auxiliary network. The frozen stable layer parameters in auxiliary network can only be updated via the unfrozen stable layers in the primary network. This method not only severs gradient interference from unstable to stable layers (via network freezing), but also enables stable layers to adapt to the stabilized parameter changes from unstable layers (via EMA).

The forward propagation processes and loss computations for both primary and auxiliary networks are as follows:

$$y^P = \tilde{PM}(x), where \quad \tilde{PM} = Cat\{PL^S, \tilde{PL}^U\}, \tag{4}$$

$$y^A = \tilde{AM}(x), where \quad \tilde{AM} = Cat\{\tilde{AL}^S, AL^U\}, \tag{5}$$

$$\Delta P\theta^S = grad\_func(y^P, \overline{y}), \quad \Delta A\theta^U = grad\_func(y^A, \overline{y}), \tag{6}$$

where $PL^S$ and $\tilde{PL}^U$ represent the unfrozen stable layers and frozen unstable layers in the primary network $\tilde{PM}$ respectively, while $\tilde{AL}^S$ and $AL^U$ correspond to the frozen stable layers and unfrozen unstable layers in the auxiliary network $\tilde{AM}$. $Cat$ denotes the concatenation of layers from the collection. $x$, $y^P$, $y^A$, and $\overline{y}$ denote the input features, predictions from primary and auxiliary networks, and the ground truth label, respectively. $grad\_func$ is the gradient computation function that acquires the gradients of stable layer parameters in the primary network $\Delta P\theta^S$ and the gradients of unstable layer parameters in the auxiliary network $\Delta A\theta^U$.

In summary, the gradients from the loss function can only update the stable layers in the primary network and the unstable layers in the auxiliary network. The frozen unstable layers in primary network and the frozen stable layers in auxiliary network are respectively updated via the EMA algorithm using the unfrozen unstable layers in auxiliary network and the unfrozen stable layers in primary network,

$$P\theta_{t+1}^S = P\theta_t^S - \Delta P\theta^S, A\theta_{t+1}^U = A\theta_t^U - \Delta A\theta^U, \tag{7}$$

$$\tilde{A}\theta_{t+1}^S = W_f \times \tilde{A}\theta_t^S + (1 - W_f) \times P\theta_{t+1}^S,$$
$$\tilde{P}\theta_{t+1}^U = W_f \times \tilde{P}\theta_t^U + (1 - W_f) \times A\theta_{t+1}^U, \tag{8}$$

where $P\theta_t^S$, $P\theta_{t+1}^S$, $A\theta_t^U$ and $A\theta_{t+1}^U$ represent the parameters of stable layers in the primary network before and after gradient updates, and the parameters of unstable layers in the auxiliary network before and after gradient updates, respectively. $\tilde{P}\theta_t^U$, $\tilde{P}\theta_{t+1}^U$, $\tilde{A}\theta_t^S$ and $\tilde{A}\theta_{t+1}^S$ represent the parameters of frozen unstable layers in the primary network before and after EMA updates, and the parameters of frozen stable layers in the auxiliary network before and after EMA updates, respectively. $W_f$ is the parameter update coefficient that controls the influence strength from corresponding layers in the another network on the current layer's parameter updates. A larger value of $W_f$ indicates weaker influence on the current layer, and it is typically set to 0.99.

## 2.4 DYNAMIC PARAMETER UPDATE

The LDT method effectively avoids gradient interference from unstable to stable layers and stabilizes unstable layers' gradient updates through its cross-freezing of stable/unstable layers and EMA algorithm. However, as shown in Figure.2b and Figure.2c, there remains significant variation in fluctuation magnitudes between layers within the unstable and stable layer groups. If the same parameter update coefficient $W_f$ is applied to all layers when EMA updating, it would inevitably result in underutilization of information and counterintuitive behavior.

When a layer exhibits higher variance (greater fluctuation magnitude), it should incorporate parameters from more timesteps to stabilize its parameter updates. Conversely, when a layer demonstrates lower variance, indicating more stable gradient updates and stronger generalization capability, its feature learning capacity should be enhanced by strengthening its parameter update efficiency.

During the EMA-based parameter update process, where frozen parameters are updated using unfrozen parameters via the EMA algorithm, a parameter update coefficient $W_f$ closer to 1 implies lower influence weights of unfrozen parameters on frozen parameters, requiring more timesteps of unfrozen parameters to induce updates to frozen parameters. Conversely, a smaller $W_f$ corresponds to higher influence weights of unfrozen parameters, enabling substantial updates to frozen parameters with fewer timesteps of unfrozen parameters. Therefore, we propose to refine the EMA algorithm's parameter updates for frozen layers by assigning larger update coefficients to high-variance frozen layers (enabling reference to more timesteps of unfrozen parameters) while giving smaller coefficients to low-variance frozen layers (allowing significant updates from fewer unfrozen parameters), thereby enhancing overall update efficiency.

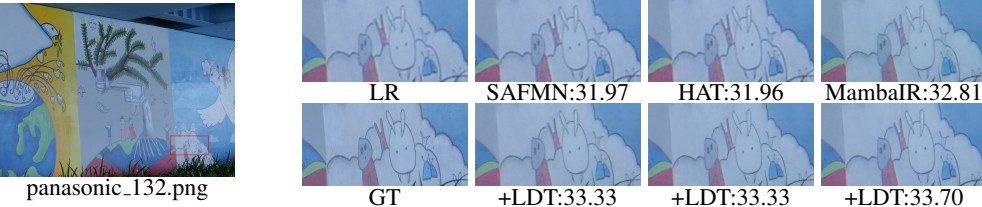

Figure 4: Visual comparison of LDT. The large image on the left is the LR image, and the sub-images on the right are LR, SAFMN, HAT, MambaIR (first row), GT, SAFMN + LDT, HAT + LDT, MambaIR + LDT (second row). The value following the name represents the PSNR metric of the current patch.

The dynamic parameter update (DPU) strategy first sorts all stable and unstable layers in descending order of their variance magnitudes, respectively, and subsequently calculates each layer's relative ranking position.

$$Rank_i^S = \frac{Get\_index(Var_i^S, Var^S)}{N^S}, where \quad i \in \{1 \cdots N^S\},$$
$$Rank_j^U = \frac{Get\_index(Var_j^U, Var^U)}{N^U}, where \quad j \in \{1 \cdots N^U\}, \tag{9}$$

where $Get\_index(Var_i^S, Var^S)$ denotes obtaining the rank order of the $i - th$ stable layer's variance $Var_i^S$ among all stable layer variances $Var^S$, and $Get\_index(Var_j^U, Var^U)$ follows the same principle. $N^S$ and $N^U$ represent the quantities of stable layers and unstable layers respectively.

Subsequently, DPU calculates the parameter update coefficient for the current layer based on its obtained ranking position:

$$W_i^S = W_{Base}^S + Rank_{Base}^S * Rank_i^S,$$
$$W_j^U = W_{Base}^U + Rank_{Base}^U * Rank_j^U, \tag{10}$$

where $W_i^S$ and $W_j^U$ denote the parameter update coefficients for the $i - th$ stable layer and $j - th$ unstable layer, respectively. $W_{Base}^S$ and $W_{Base}^U$ are set to 0.99 and 0.999 respectively, while $Rank_{Base}^S$ and $Rank_{Base}^U$ are configured as 0.01 and 0.001.

Referring to Eq.8, the frozen unstable layers in the primary network and the frozen stable layers in the auxiliary network are updated using their corresponding parameter update coefficients,

$$\tilde{A}\theta_{t+1}^S = W^S \times \tilde{A}\theta_t^S + (1 - W^S) \times P\theta_{t+1}^S,$$
$$\tilde{P}\theta_{t+1}^U = W^U \times \tilde{P}\theta_t^U + (1 - W^U) \times A\theta_{t+1}^U, \tag{11}$$

where $W^S \in \{W_1^S, \cdots, W_{N^S}^S\}$, $W^U \in \{W_1^U, \cdots, W_{N^U}^U\}$ represent the parameter update coefficients for stable layers in the primary network and unstable layers in the auxiliary network respectively.

## 2.5 TEST-TIME INFERENCE

After training the dual-branch network using LDT and DPU strategies, we extract (1) the frozen unstable layers from the primary network and (2) the frozen stable layers from the auxiliary network, then concatenate them to construct the composite network $MC$. During test-time inference, $MC$ performs predictions on input samples. This workflow can be formally expressed as:

$$MC = Cat\{\tilde{A}L^S, \tilde{P}L^U\}, \tag{12}$$
$$y = MC(x), \tag{13}$$

where $x$ and $y$ denote the input sample and its corresponding prediction, respectively.

Table 2: Effectiveness validation of LDT. The samples from the Olympus-camera branch are selected as the source domain, while those from the remaining camera branches constitute the target domain. Performance is evaluated using PSNR and SSIM metrics. **The experiment was repeated three times, with results reported as mean $\pm$ standard deviation.**

| Method | Pan | Sony | DSC |
|---|---|---|---|
| Baseline | 30.81/0.8688 | 30.81/0.8850 | 30.22/0.8753 |
| LDT | $31.20 \pm 0.0883$/$0.8631 \pm 4.58$e-4 | $31.25 \pm 0.2768$/$0.8746 \pm 4.79$e-3 | $31.23 \pm 0.0923$/$0.8869 \pm 7.02$e-4 |
| LDT & DPU | $31.36 \pm 0.0469$/$0.8611 \pm 6.03$e-4 | $32.15 \pm 0.1953$/$0.8880 \pm 3.03$e-3 | $31.51 \pm 0.1351$/$0.8865 \pm 1.16$e-3 |

| Method | IMG | Canon | |
|---|---|---|---|
| Baseline | 30.01/0.8737 | 30.93/0.8617 | |
| LDT | $30.17 \pm 0.1989$/$0.8730 \pm 5.69$e-4 | $32.33 \pm 0.2187$/$0.9236 \pm 1.23$e-3 | |
| LDT & DPU | $30.57 \pm 0.1367$/$0.8705 \pm 3.61$e-4 | $32.80 \pm 0.2560$/$0.9246 \pm 1.51$e-3 | |

Table 3: Ablation experiments on stable/unstable layer partitioning criteria

| Method | Pan | Sony | DSC | IMG | Canon |
|---|---|---|---|---|---|
| Baseline | 30.81/**0.8688** | 30.81/0.8850 | 30.22/0.8753 | 30.01/**0.8737** | 30.93/0.8617 |
| Random | 30.96/0.8619 | 30.88/0.8692 | 31.00/0.8858 | 30.02/0.8732 | 32.05/0.9217 |
| Mean | 31.18/0.8598 | 31.86/0.8833 | 31.28/0.8854 | 30.47/0.8706 | 32.39/0.9226 |
| Var/Mean | 31.27/0.8615 | 31.89/0.8849 | 31.37/**0.8870** | 30.50/0.8717 | 32.59/**0.9248** |
| Var | **31.36**/0.8611 | **32.15**/**0.8880** | **31.51**/0.8865 | **30.57**/0.8705 | **32.80**/0.9246 |

## 3 EXPERIMENTS

### 3.1 EXPERIMENTAL SETUP

**Datasets:** For image SR tasks, we employ the DRealSR dataset (Wei et al., 2020), which consists of images captured by multiple cameras (Olympus, Pan, Sony, DSC, IMG, Canon). Since each camera possesses distinct hardware parameters, samples collected by different cameras exhibit significant domain shift. During experiments, we select images from one or multiple cameras as the source domain, while using images from the remaining cameras as the target domain. For image classification tasks, we employ the VLCS dataset (Torralba & Efros, 2011), which comprises the VOC, LabelMe, Caltech, and SUN datasets. The VOC dataset contains diverse daily-life scene images, the LabelMe dataset exhibits multi-scene characteristics, the Caltech dataset focuses on specific objects (e.g., vehicles), and the SUN dataset covers various indoor and outdoor scene images. We adopt the DomainBed-consistent training strategy, specifically cross-training validation. For semantic segmentation tasks, we employ the Cityscapes (Cordts et al., 2016), BDD100K (Yu et al., 2018), and Mapillary (Neuhold et al., 2017) datasets, which contain diverse autonomous driving scenarios with distinct styles.

**Network architecture:** For SR tasks, we validate the effectiveness of our proposed method on networks based on CNN, Transformer, and the recently popular Mamba architectures, specifically SAFMN (Sun et al., 2023), HAT (Chen et al., 2023), and MambaIR (Guo et al., 2024a) respectively. For the image classification task, we employ ResNet-18, ResNet-50(He et al., 2016a), ViT (Dosovitskiy et al., 2021), and Vision Mamba (Liu et al., 2024) network architectures. For semantic segmentation tasks, we adopt an architecture consisting of a ResNet-50 He et al. (2016b) backbone with a DeepLabV3+ Chen et al. (2018) prediction head.

**Implementation details:**

The input patch sizes are $48 \times 48$ for SR tasks, $224 \times 224$ for classification tasks, and $512 \times 512$ for semantic segmentation tasks. We employ $4\times$ V100 GPUs as training devices. It is worth noting that since SR data processing is relatively straightforward and constitutes a pixel-level task, it is more susceptible to domain shift effects. Consequently, we conducted ablation experiments on the SR task branch.

Table 4: Ablation experiments on training/inference efficiency. The task is image super-resolution, with training and inference patch sizes set to 48×48 and 200×200 pixels respectively. The network architecture is based on MambaIR.

| Method | Training memory (GB) | Inf memory (GB) | Training time (s) | Inf time (s) |
|---|---|---|---|---|
| Baseline | 15.27 | 2.7 | 0.6912 | 658.3287 |
| DeFT (Pahk et al., 2025) | 20.30 | 2.7 | 1.2566 | 653.9900 |
| LDT | 20.25 | 2.7 | 1.2608 | 643.6736 |

Table 5: Comparative experiments

| Network | Pan | Sony | DSC | IMG | Canon |
|---|---|---|---|---|---|
| IODA (Tang & Yang, 2024) | 30.97/0.8594 | 31.31/0.8807 | 31.05/0.8852 | 30.15/0.8728 | 31.87/0.9216 |
| SRTTA (Deng et al., 2023) | 29.88/0.8359 | 31.24/0.8714 | 29.93/0.8639 | 29.78/0.8580 | 31.88/0.9146 |
| Wang et al. (2024a) | 31.28/0.8626 | 31.53/0.8818 | 31.34/0.8875 | 30.42/**0.8775** | 32.72/**0.9269** |
| DTAM (Huang et al., 2024) | 31.23/0.8615 | 31.29/0.8773 | 31.29/0.8864 | 30.32/0.8747 | 32.65/0.9256 |
| START (Guo et al., 2024b) | 31.28/0.8609 | 31.41/0.8774 | 31.29/0.8862 | 30.33/0.8743 | 32.70/0.9261 |
| MambaIR + LP-FT (Kumar et al., 2022) | 30.99/0.8621 | 30.97/0.8722 | 30.82/0.8827 | 29.93/0.8696 | 31.64/0.9212 |
| MambaIR + DeFT (Pahk et al., 2025) | 31.27/**0.8632** | 31.61/0.8801 | 31.34/**0.8875** | 30.31/0.8726 | 32.40/0.9247 |
| MambaIR + LDT | **31.36**/0.8611 | **32.15**/**0.8880** | **31.51**/0.8865 | **30.57**/0.8705 | **32.80**/0.9246 |

## 3.2 ABLATION EXPERIMENTS

### 3.2.1 ABLATION EXPERIMENTS FOR EACH COMPONENT OF LDT

To validate the effectiveness of our proposed method, we conduct ablation studies for each module. The performance metrics of models fine-tuned on the source domain and evaluated on the target domain serve as baseline results.

We first evaluate the network performance with only the Layer-Decomposition Training (LDT) strategy implemented (without DPU). As shown in Table. 2, LDT improves the SR network's performance across all target-domain camera branches, with the most significant PSNR gain of 1.4 dB observed on the Canon data branch. By isolating gradients between stable and unstable layers, the LDT strategy prevents perturbations from large parameter fluctuations in unstable layers during stable layer updates, effectively enhancing parameter update stability. Subsequently, we incorporate the proposed Dynamic Parameter Update (DPU) strategy with LDT, yielding further performance improvements on the target domain - notably a 0.4dB PSNR increase on the Sony branch. Through finer-grained processing of gradient amplitude variations within both unstable and stable layers, DPU enhances update adaptability and further boosts the network's generalization capability.

### 3.2.2 ABLATION EXPERIMENTS ON STABLE/UNSTABLE LAYER PARTITIONING CRITERIA

To verify the impact of stable/unstable layer partitioning criteria on network generalization performance, we conducted the following experiments: (1) Random partition, where layers were randomly assigned as stable or unstable. (2) Gradient magnitude-based partition, where layers were sorted by their mean gradient magnitudes across input samples, with layers exhibiting larger mean gradients designated as unstable. (3) Variance-based partition, where layers were sorted by gradient variance across input samples, assigning those with higher variance as unstable. (4) Normalized gradient variance partition. Under identical fluctuation amplitudes, layers with higher gradients exhibit greater variance than those with lower gradients. To eliminate this bias, we implement normalized gradient variance partitioning (Var/Mean).

As shown in Table.3, random partitioning of stable and unstable layers provides only marginally improvements in network generalization performance. While the dual-branch training strategy enhances parameter update stability via EMA, misclassification between unstable and stable layers reduces gradient interference isolation efficiency, ultimately limiting generalization gains. Using mean gradient magnitude for stable/unstable layer partitioning yields modest generalization improvements, though underperforms LDT's variance-based method. Large gradients primarily emerge from two scenarios: (1) the network encountering new distribution samples requiring adaptation, and (2) certain layers exhibiting excessive sensitivity to input variations. Employing only gradient averages for layer separation would incorrectly categorize the first scenario's layers as unstable (subsequently frozen during fine-tuning), thereby impairing the network's feature learning capacity. As mentioned in Section 2.3.2, the first scenario produces more coherent parameter updates with lower

gradient variance, owing to the aligned distribution of training samples, whereas the second scenario demonstrates more randomized update directions and larger gradient variance. Using variance as the metric to distinguish stable and unstable layers effectively distinguishes between these two scenarios, achieving strong performance across all four camera branches in the target domain. While normalized variance outperforms variance in certain camera branches, the LDT method adopts variance as the layer partition metric to preserve methodological simplicity.

### 3.3 ABLATION EXPERIMENTS ON TRAINING/INFERENCE EFFICIENCY

To validate the method's impact on computational efficiency, we systematically evaluate GPU memory consumption during both the training and inference stages for: (1) the baseline method, (2) DeFT [32], and (3) our proposed LDT, as quantified in Table 4. We further measure the training time per image and inference time across the entire Olympus-camera branch dataset for all compared methods.

Although DeFT and LDT introduce an auxiliary network, their additional parameters remain frozen (excluding them from gradient computation), resulting in limited memory overhead. Crucially, during inference, LDT maintains identical memory consumption to the baseline since only a single network processes input images.

### 3.4 COMPARATIVE EXPERIMENTS

As shown in Table.5, we compare our proposed LDT method with other domain generalization methods, including parameter-correlation-focused methods LP-FT (Kumar et al., 2022) and DeFT (Pahk et al., 2025), feature-perturbation-based domain generalization methods START (Guo et al., 2024b), DTAM(Huang et al., 2024), and Wang et al. (2024a), as well as domain adaptation methods IODA (Tang & Yang, 2024) and SRTTA (Deng et al., 2023) trained on both source and target domains. For instance, compared to DeFT, LDT obtains a PSNR improvement of 0.54 dB on the Sony data branch. Visual comparisons in Appendix Figure F further confirm that the super-resolved images produced by LDT exhibit clearer and more accurate texture details.

## 4 CONCLUSION

In this paper, we propose the Layer-Decomposition Training (LDT) strategy, which effectively mitigates feature distribution perturbations caused by misclassified unstable layers in existing methods through layer-wise separation of stable and unstable layers. Furthermore, the proposed Dynamic Parameter Update (DPU) strategy enhances the network's adaptability to amplitude variations within both stable and unstable layers by adaptively determining update coefficients based on gradient oscillation levels, thereby improving generalization performance. Extensive experiments across diverse tasks and architectures demonstrate LDT's effectiveness and universality.

## ACKNOWLEDGMENTS

This work was supported by the Fundamental and Interdisciplinary Disciplines Breakthrough Plan of the Ministry of Education of China (Grant JYB2025XDXM118), and the Natural Science Foundation of China (Grant 62176119). Additionally, we would like to thank Wei-Wei Zhao for conducting the LDT experiments in the NLP classification tasks.

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

## A  RELATED WORK

### A.1  DATA AUGMENTATION-BASED DOMAIN GENERALIZATION METHODS

Data augmentation strategies (Vaish et al., 2024a; Cao et al., 2022; Wang et al., 2021; Castellini et al., 2023; Xu et al., 2022) aimed to reduce neural networks' overfitting problems through data perturbation and increasing sample feature diversity, thereby improving the networks' generalization performance.

For image classification tasks, the Mixup (Zhang et al., 2017) data augmentation method randomly mixed two images while proportionally transforming the class labels according to the mixing ratio, effectively increasing the diversity of original samples. CutMix (Yun et al., 2019) randomly cropped partial regions of images and covered the cropped regions with processed crops from other images. Liu et al. (2022) designed a token-level Mix strategy specifically for Transformer networks. Islam et al. (2024) argued that randomly mixing two images may not only omit important portions of the input images but also introduce label ambiguities. Therefore, they employed a Diffusion architecture for image generation, avoiding the label ambiguity problem. Fan et al. (2024) designed a data augmentation strategy for instance segmentation tasks, effectively expanding the diversity of training samples. Wang et al. (2024c) proposed a foreground-background separation-based data augmentation strategy, where they effectively enhanced training sample diversity by combining foreground features with different background features. Feng et al. (2019) addressed the overfitting problem of SR models with small data samples by proposing a Mixup data augmentation strategy, which directly merged two LR images at random ratios, effectively increasing training sample diversity. Yoo et al. (2020) argued that existing data augmentation strategies like CutMix could destroy spatial relationships between image pixels, harming SR task performance. Therefore, they adopted a cross-augmentation method (CutBlur) that pasted HR images onto upsampled LR images and added upsampled LR images to HR images. They claimed that compared to existing data augmentation methods, CutBlur not only taught the network how to perform SR but also taught it which regions to super-resolve, preventing the network from producing overly sharp images. Xiao et al. (2023) first proposed a data augmentation strategy specifically for light field image SR tasks. They randomly augmented images using CutBlur on light field images' unique multi-view images, effectively improving light field SR performance. Chao et al. (2024) further enhanced light field SR performance by proposing a dual spatial-angular data augmentation strategy based on CutMix. The CutBlur (Mi & Yang, 2025) method achieved sample diversity expansion by mutually pasting and covering LR and HR images. However, the ADD method (Zeyu & Yubin, 2025) discovered that CutBlur could cover high-information regions in images, causing information loss. To address this, they introduced an attribution algorithm to guide the pasting process, ensuring only low-information regions were covered each time, effectively preserving information richness.

## A.2 Domain-invariant Feature Learning-based Domain Generalization Methods

Domain generalization methods targeting domain-invariant features typically decomposed input features into domain-invariant and domain-specific components. By preserving learning capacity for domain-invariant features while reducing sensitivity to domain-specific variations, these methods effectively enhanced model generalization performance.

Previous domain generalization methods primarily focused on processing either high-level or low-level features. DomainDrop (Guo et al., 2023), however, operated along the channel dimension by identifying and suppressing channels containing domain-specific features through channel activation values. The Dropout method (Hinton et al., 2012) improved network robustness by randomly dropping connections between layers. However, Wang et al. (2024a) argued that using Dropout in low-level vision tasks could lead to loss of feature diversity, thereby degrading network performance. Given this limitation, they proposed a degradation consistency loss that enforced consistent predictions across differently degraded images, thus enhancing network robustness. Ahn et al. (2024) maintained that image augmentation should not alter the relationships between objects in images. They constrained objects in augmented images using covariance and employed contrastive learning to enhance feature discriminability while preserving generalization performance. Chattopadhyay et al. (2023) investigated domain generalization from virtual to real scenes, noting that "synthetic images have less variance in high-frequency components of the amplitude spectra compared to real images." Based on this assumption, PASTA perturbed the amplitude spectra of synthetic images in the fourier domain to generate augmented views. DGMamba (Long et al., 2024) conducted domain generalization research on the recently popular Mamba network architecture (Gu & Dao) . They addressed the issue of accumulated domain shift caused by iterative hidden state updates in Mamba networks by proposing a hidden state random shuffling data augmentation strategy. The START method (Guo et al., 2024b) further refined input feature processing by dividing features into foreground features (affecting predictions) and domain-specific background features based on activation values. START improved background robustness through style swapping between background features and randomly generated features. Huang et al. (2024) improved the foreground/background

feature partitioning strategy. They argued that foreground features determining object predictions should exhibit high correlation with other patch features. Thus, they identified patches with high covariance values as foreground features and others as background. Li et al. (2024a) introduced CLIP to domain generalization tasks, using text descriptions with CLIP's text encoder to guide feature learning, and implemented domain-specific feature filtering through generated channel and spatial masks, effectively improving network robustness. Cheng et al. (2024) leveraged large language models to reason about domain-specific and domain-invariant features, constructing a memory bank from domain-specific features to guide subsequent inference. Zhao et al. (2022) proposed a test-time domain generalization method, hypothesizing that amplitude images from fourier transforms contained feature intensity information (considered as style information and domain-specific). They therefore augmented amplitude images to enhance robustness and reduced feature distance between test and source domain samples by incorporating source domain features during testing. Park et al. (2023) further categorized test samples, retaining those similar to source domain features while applying style transfer to samples with large distribution gaps. Yu & Hwang (2024) added noise prompts around input images to reduce distribution distance between test and source domain samples.

Existing domain generalization methods primarily investigated three directions: input samples, intermediate features, and network architectures, while largely neglecting parameter correlation analysis. Due to gradient backpropagation through layers, parameter updates exhibited strong interdependencies, where fluctuations in individual parameters induced network-wide perturbations that significantly degraded generalization performance.

Kumar et al. (2022) discovered that linear probing demonstrated superior performance compared to full fine-tuning when handling samples with large domain shifts, while full fine-tuning outperformed linear probing on data with smaller distribution shifts. Therefore, they divided the training process into two stages: first initializing the prediction head using linear probing, then adjusting the entire network through full fine-tuning, effectively enhancing network robustness. Pahk et al. (2025) observed that backbone networks pre-trained on large datasets demonstrated superior feature extraction capabilities compared to randomly initialized prediction heads. They demonstrated that joint fine-tuning of both components enabled the prediction head to perturb backbone features, consequently degrading its representational capacity. To mitigate this, they decoupled the fine-tuning processes of the backbone and prediction head, while introducing a parallel auxiliary network to stabilize parameter updates in both components.

The method of distinguishing stable and unstable parameters simply by separating backbone network and prediction head was naive. As shown in Figure 2, during network training, certain layers in the backbone network exhibited greater instability compared to the prediction head, which could impair the backbone's feature processing capability. Furthermore, existing parameter update strategies applied identical weight coefficients to layers with different stability levels, showing limited adaptability. To address these issues, we proposed Layer-decomposition Training (LDT), which further reduced feature corruption from unstable layers through finer-grained layer decomposition. We also introduced the Dynamic Parameter Update (DPU) strategy that adaptively adjusted update weights according to each layer's stability characteristics, achieving superior adaptability.

## B  THEORETICAL PROOFS

Following (Kumar et al., 2022; Pahk et al., 2025), for clarity of explanation, we simplify the network into two modules, $S$ and $U$, where $S$ denotes stable layers, which exhibit more stable parameter updates (lower gradient variance), and $U$ denotes Unstable layers, which exhibit unstable parameter updates (higher gradient variance).

**Theorem 1** A network comprises one stable layer $S$ and one unstable layer $U$. At time step $t + 1$ of network parameter updates, the gradient $\Delta S_{t+1}$ of stable layers $S$ shows strong correlation with the gradient $\Delta U_t$ of unstable layers $U$ at time step $t$. This can be expressed as $\Delta S_{t+1} = f(\Delta S_t, \Delta U_t, x)$, where $\Delta S_t$ denotes the gradient of stable layers $S$ at step $t$, and $x$ represents the input data.

**Proof.** At time step $t + 1$, the network's prediction can be expressed as:

$$y_{t+1} = (S_t - \Delta S_t)(U_t - \Delta U_t)x. \tag{14}$$

When the loss function is the L1 loss adopted in super-resolution tasks, i.e., $Loss_{L1} = y_{t+1} - \overline{y}$, where $\overline{y}$ denotes the ground truth label. At time step $t+1$, the gradient of module $S$ can be expressed as:

$$\Delta S_{t+1}^{L1} = (1 - \frac{\partial \Delta S_t}{\partial S_t})(U_t - \Delta U_t)x + (S_t - \Delta S_t)(-\frac{\partial \Delta U_t}{\partial \Delta S_t}\frac{\partial \Delta S_t}{\partial S_t}). \quad (15)$$

From Eq.15, it can be observed that at time step $t + 1$, the gradient of module $S$ is influenced by the gradient $\Delta U_t$ of module $U$ at time $t$. Consequently, the instability characteristics in unstable layers ultimately affect parameter updates in stable layers, thereby interfering with the overall network's predictive performance. Eq.15 can be simplified as $\Delta S_{t+1}^{L1} = f^{L1}(\Delta S_t, \Delta U_t, x)$.

Assuming the loss function is the L2 loss commonly employed in high-level vision tasks, i.e., $Loss_{L2} = (y_{t+1} - \overline{y})^2$, the gradient of module $S$ at time step $t + 1$ can be expressed as:

$$\Delta S_{t+1}^{L2} = 2[(S_t - \Delta S_t)(U_t - \Delta U_t)x - \overline{y}] * f^{L1}(\Delta S_t, \Delta U_t, x), \quad (16)$$

where $f^{L1}(\Delta S_t, \Delta U_t, x)$ represents the gradient update magnitude at time step $t + 1$ under the L1 loss.

From Eq. 16, it can similarly be observed that at time step $t + 1$, the gradient update of module $S$ remains correlated with the gradient update of module $U$ at time $t$.

When the loss function is the cross-entropy employed for classification tasks, i.e., $Loss_{Cross} = \overline{y}log(y)$, the gradient of module $S$ at time step $t + 1$ can be expressed as:

$$\Delta S_{t+1}^{Cross} = \overline{y}\frac{f^{L1}(\Delta S_t, \Delta U_t, x)}{(S_t - \Delta S_t)(U_t - \Delta U_t)x}. \quad (17)$$

From Eq. 17, it can similarly be observed that in classification tasks, the gradient update of module $S$ at time step $t + 1$ is also influenced by the gradient of module $U$ at time $t$.

Through Eq. 15, 16, and 17, it can be observed that the gradient updates of unstable layers $U$ significantly influence subsequent gradient updates of stable layers $S$ parameters. Therefore, the gradient of stable layers $S$ at time step $t + 1$ can be expressed as $\Delta S_{t+1} = f(\Delta S_t, \Delta U_t, x)$, where $\Delta S_t$ denotes the gradient of stable layers $S$ at step $t$, and $x$ represents the input data.

**Extension to Multi-Layer Network**

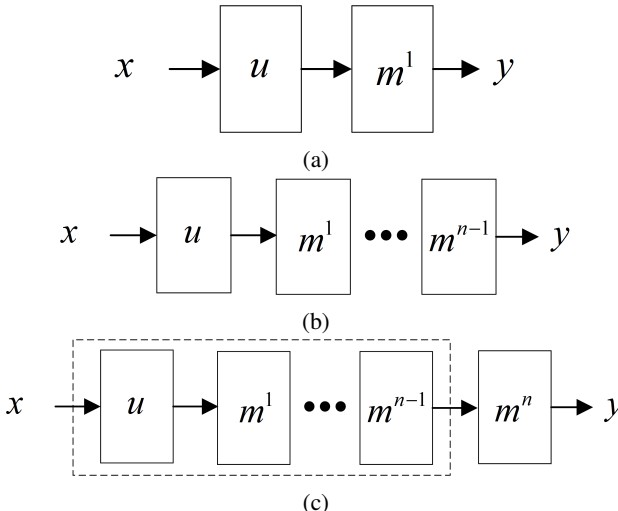

Figure 5: Network architectures under different scenarios. $u$ denotes an unstable layer, while $m$ represents an unspecified layer that could be either stable or unstable. $x$ and $y$ are the network input and output, respectively.

**Theorem 2** For a network comprising one $U$ layer and $k$ unspecified layers $\{m^1, ..., m^k\}$, where each $m$ could be either stable or unstable, the gradients of all $k$ unspecified layers $m$ are influenced by the gradients of the unstable $U$ layer, as expressed by $\Delta(m^1 m^2 \ldots m^k) = f(\Delta U, ...)$.

**Proof.** (We proceed by mathematical induction on $k$.)

1. Base Case ($k = 1$):

As show in Figure 5a, when the network consists of an unstable layer $U$ and an unspecified layer $m^1$, the gradient of $m^1$ at time $t + 1$, denoted as $\Delta m^1_{t+1}$, is influenced by the gradient of $U$ at time $t$, $\Delta U_t$. (This relationship is analogous to Theorem 1, where the stable layer $S$ is replaced by the unspecified layer $m^1$.) This relationship is expressed as $\Delta m^1_{t+1} = f^1(\Delta U_t, ...)$. It should be noted that, unlike in Theorem 1, the function $f(\cdot)$ in this context represents a relationship between two variables, with the former being influenced by the latter.

2. Inductive Hypothesis ($k = n - 1$):

As shown in Figure 5b, assume that for $k = n - 1$, $U$ forms a combined unit with $n - 1$ unspecified layers $\{m^1 \dots m^{n-1}\}$. The gradient of the combined unit $m^1 m^2 \dots m^{n-1}$ depends on the gradient of $U$, satisfying $\Delta(m^1 m^2 \dots m^{n-1}) = f^{n-1}(\Delta U, ...)$.

3. Inductive Step ($k = n$):

As shown in Figure 5c, for the case of $k = n$, the network comprising one $U$ layer and $n$ unspecified layers $\{m^1, \dots, m^n\}$ is constructed by adding an $m^n$ layer to the architecture with $k = n - 1$. The network output at time $t + 1$ can be expressed as:

$$y_{t+1} = (Um^1 m^2 \dots m^{n-1} - \Delta(Um^1 m^2 \dots m^{n-1})_t)(m^n - \Delta m^n_t)x \tag{18}$$

The gradient of $m^n$ can be expressed as:

$$\begin{aligned} \Delta m^n_{t+1} =& (-\frac{\partial \Delta(Um^1 m^2 \dots m^{n-1})_t}{\partial m^n})(m^n - \Delta m^n_t)x \\ &+ (Um^1 m^2 \dots m^{n-1} - \Delta(Um^1 m^2 \dots m^{n-1})_t)(1 - \frac{\partial \Delta m^n_t}{\partial m^n})x \end{aligned} \tag{19}$$

It can be abbreviated as:

$$\Delta m^n_{t+1} = f^n(\Delta(Um^1 m^2 \dots m^{n-1})_t, \Delta m^n_t, x) \tag{20}$$

Given that $\Delta(m^1 m^2 \dots m^{n-1}) = f^{n-1}(\Delta U, ...)$, it follows that $\Delta(Um^1 m^2 \dots m^{n-1}) = f^{Un-1}(\Delta U, ...)$. Substituting this into Eq. 20, we can further deduce that $\Delta m^n_{t+1}$ depends on $\Delta U$. Consequently, the gradient of the combined unit $m^1 m^2 \dots m^{n-1} m^n$ also depends on the gradient of $u$, as expressed by $\Delta(m^1 m^2 \dots m^n) = f^n(\Delta U, ...)$.

4. Conclusion:

For a network comprising one $U$ layer and $k$ unspecified layers $\{m^1, ..., m^k\}$, we conclude that for any $k \geq 1$, the gradients of all $k$ unspecified layers in such a network are influenced by the gradients of the unstable layer $U$, as expressed by $\Delta(m^1 m^2 \dots m^k) = f(\Delta U, ...)$.

## C  THE SINGLE-BRANCH VERSION OF LDT (LDT-S)

### C.1  METHOD

The LDT method, which computes gradients for both the primary and auxiliary networks, involves notable computational overhead. To improve its efficiency, we introduce LDT-S, a streamlined single-branch version. LDT-S achieves lower computational cost by alternately freezing parts of the single network across timesteps, maintaining the requisite isolation between stable and unstable layers while ensuring the stabilization of the unstable ones.

As shown in Figure 6, the intermediate $2 \times ss$ timesteps of LDT-S training serve as an example. This example consists of two stages: the unstable layers training stage and the stable layers training stage, with each stage spanning $ss$ timesteps. The entire LDT-S training process employs only a single network and one weight buffer. This buffer, located in the CPU, is used to store the network weights from the immediately preceding timestep only.

The unstable layers training stage is initiated by loading the network weights from the preceding timestep stored in the weight buffer. To prevent the unstable layers from interfering with the training

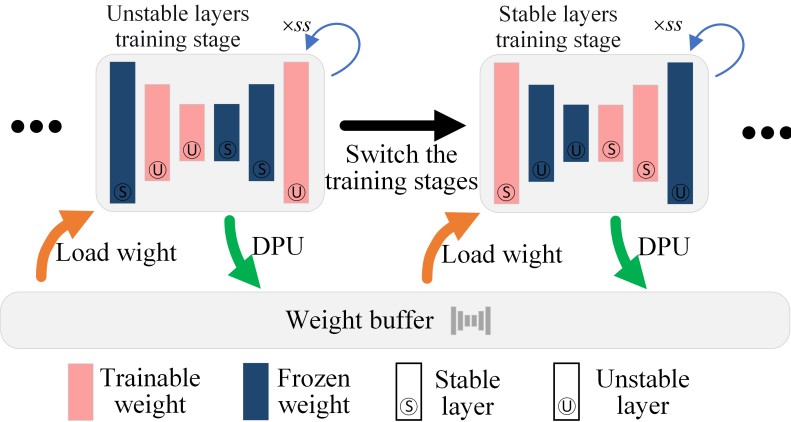

Figure 6: The overall framework of LDT-S. $ss$ is the side-switching interval, determining the number of training timesteps allocated to each stage.

of the stable layers, the parameters of the stable layers are frozen in this stage, and gradients are computed and used to update only the unstable layers. During this stage, the unstable layer parameters in the network are used to update their counterparts in the weight buffer via the DPU method, thereby stabilizing the unstable layer parameters stored in the weight buffer. After this stage executes for $ss$ timesteps, a training stage switching occurs, transitioning the process into the stable layers training stage.

The stable layers training stage analogously begins by loading the weights from the previous timestep from the weight buffer and proceeds to freeze the unstable layers. This stage utilizes the DPU method to update the stable layer parameters stored in the weight buffer, enabling them to adapt to the newly stabilized unstable layers. After $ss$ timesteps, a training stage switching is executed again, transitioning back to the unstable layers training stage. Finally, upon the completion of network training, the final weights are retrieved from the weight buffer for inference and testing.

The pseudocode for the overall training procedure of LDT-S is provided in Algorithm 3.

## C.2 EXPERIMENTS ON PERFORMANCE AND EFFICIENCY

As shown in Table 6, owing to its single-branch design, LDT-S exhibits superior computational efficiency compared to LDT, with an efficiency nearly on par with the single-network baseline model. Furthermore, LDT-S achieves competitive performance against the dual-branch method DeFT (Table 7), while also attaining superior computational efficiency.

Table 6: Ablation experiments on training/inference efficiency. The task is image super-resolution, with training and inference patch sizes set to $48 \times 48$ and $200 \times 200$ pixels respectively. The network architecture is based on MambaIR.

| Method | Training memory (GB) | Inference memory (GB) | Training time (s) | Inference time (s) |
|---|---|---|---|---|
| Baseline | 15.27 | 2.7 | 0.6912 | 658.3287 |
| DeFT | 20.30 | 2.7 | 1.2566 | 653.9900 |
| LDT-S | 15.21 | 2.7 | 0.6920 | 649.5462 |
| LDT | 20.25 | 2.7 | 1.2608 | 643.6736 |

Table 7: Performance comparison

| Method | Pan | Sony | DSC | IMG | Canon |
|---|---|---|---|---|---|
| Baseline | 30.81/0.8688 | 30.81/0.8850 | 30.22/0.8753 | 30.01/0.8737 | 30.93/0.8617 |
| DeFT | 31.27/0.8632 | 31.61/0.8801 | 31.34/0.8875 | 30.31/0.8726 | 32.40/0.9247 |
| LDT-S | 31.30/0.8600 | 31.46/0.8793 | 31.37/0.8878 | 30.37/0.8729 | 32.49/0.9228 |
| LDT | 31.36/0.8611 | 32.15/0.8880 | 31.51/0.8865 | 30.57/0.8705 | 32.80/0.9246 |

# D ADDITIONAL ABLATION EXPERIMENTS

## D.1 ABLATION EXPERIMENTS ON LAYER PARTITIONING STRATEGIES

As shown in Table 8, we performed an ablation study on layer partitioning strategies, comparing a static strategy, which maintains the initial layer assignment throughout training, with a dynamic strategy that reassigns stable and unstable layers every 5,000 steps. The dynamic partitioning strategy achieves higher PSNR and SSIM scores than the static strategy across the Pan, DSC, and Canon data branches. This improvement stems from its ability to better adapt to the training process by dynamically identifying stable and unstable layers based on evolving parameter states, demonstrating superior adaptability.

Table 8: Ablation experiments on layer partitioning strategies

| Method | Pan | Sony | DSC | IMG | Canon |
|---|---|---|---|---|---|
| Baseline | 30.81/0.8688 | 30.81/0.8850 | 30.22/0.8753 | 30.01/0.8737 | 30.93/0.8617 |
| Static | 31.36/0.8611 | 32.15/0.8880 | 31.51/0.8865 | 30.57/0.8705 | 32.80/0.9246 |
| Dynamic | 31.41/0.8636 | 31.85/0.8826 | 31.52/0.8892 | 30.49/0.8729 | 32.76/0.9250 |

## D.2 ABLATION EXPERIMENTS ON DIFFERENT WEIGHT COEFFICIENT $W$ PROJECTION METHODS

As shown in Table 9, we compared two strategies for generating weight coefficients $W$: one directly projects normalized variance into weight coefficients $W$, and the other generates $W$ based on variance ranking. Generating weights directly from normalized variance requires careful design of the projection function to ensure stable performance. In contrast, the ranking-based method for setting weight coefficients is more straightforward. As visualized in Figure 7, we present a plot of the variance against the corresponding weight coefficients generated by the ranking-based method.

Table 9: Ablation Study on Different Weight Coefficient W Projection Methods

| Method | Pan | Sony | DSC | IMG | Canon |
|---|---|---|---|---|---|
| Baseline | 30.81/0.8688 | 30.81/0.8850 | 30.22/0.8753 | 30.01/0.8737 | 30.93/0.8617 |
| Normalized variance | 31.09/0.8624 | 31.01/0.8702 | 30.05/0.8726 | 31.07/0.8855 | 32.31/0.9227 |
| Ours | 31.36/0.8611 | 32.15/0.8880 | 31.51/0.8865 | 30.57/0.8705 | 32.80/0.9246 |

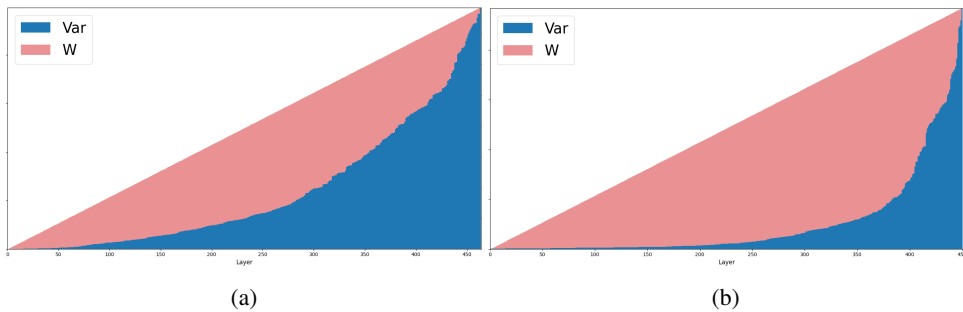

Figure 7: Correspondence between Weight Coefficients and Variance. (a) For stable layers. (b) For unstable layers.

## D.3 ABLATION EXPERIMENTS ON DIFFERENT SOURCE DOMAIN SAMPLE DISTRIBUTIONS

As shown in Table.10, to validate the robustness of the LDT method across varying source domain sample distributions, we individually designated the Olympus and Pan camera branches as the source domain. Additionally, we conducted experiments with multiple source domains by forming pairwise combinations of two out of the three data branches (Olympus, Pan and Sony) as the source domain.

Table 10: Ablation experiments on different source domain sample distributions. We employ the abbreviations O for Olympus, Pa for Pan, S for Sony, D for DSC, I for IMG, and C for Canon. Olympus + FT, S + FT, and Pa + FT denote the performance of the network on the target-domain camera branches after naive fine-tuning on the Olympus, Sony, and Pan camera branches respectively, which we use as baselines.

| Source | T1 | T2 | T3 | T4 | T5 |
|---|---|---|---|---|---|
| Olympus + FT | Pa:30.81/0.8688 | S:30.81/0.8850 | D:30.22/0.8753 | I:30.01/0.8737 | C:30.93/0.8617 |
| Olympus + LDT | Pa:31.36/0.8611 | S:32.15/0.8880 | D:31.51/0.8865 | I:30.57/0.8705 | C:32.80/0.9246 |
| Pa+ FT | Olympus:30.65/0.8596 | S:31.59/0.8854 | D:31.50/0.8906 | I:30.04/0.8715 | C:32.50/0.9262 |
| Pa +LDT | Olympus:30.65/0.8569 | S:31.76/0.8856 | D:31.57/0.8890 | I:30.20/0.8693 | C:32.51/0.9238 |
| Olympus + FT | Pa:30.81/0.8688 | S:30.81/0.8850 | D:30.22/0.8753 | I:30.01/0.8737 | C:30.93/0.8617 |
| Olympus + Pa + LDT | - | S:31.95/0.8862 | D:31.63/0.8915 | I:30.38/0.8737 | C:32.92/0.9260 |
| Olympus + S + LDT | Pa:31.41/0.8648 | - | D:31.32/0.8875 | I:30.41/0.8751 | C:32.40/0.9225 |
| S + FT | Olympus:30.69/0.8528 | Pa:31.27/0.8621 | D:31.35/0.8842 | I:30.30/0.8720 | C:32.48/0.9223 |
| S + Pa + LDT | Olympus:30.81/0.8574 | - | D:31.55/0.8872 | I:30.35/0.8709 | C:32.65/0.9239 |

## D.4 ABLATION EXPERIMENTS ON DIFFERENT NETWORK ARCHITECTURES

To validate the robustness of the LDT method across different network architectures, we conduct performance evaluations on three distinct frameworks: the CNN-based SAFMN (Sun et al., 2023), the Transformer-based HAT (Chen et al., 2023), and the recently popular Mamba-based MambaIR (Guo et al., 2024a) networks. As demonstrated in Table.11, LDT achieves consistent generalization improvements across all architectures, with the most significant performance gain observed on the Canon camera branch of MambaIR network, where the PSNR metric improves by 1.61dB.

Table 11: Ablation experiments on different network architectures

| Network | Pan | Sony | DSC | IMG | Canon |
|---|---|---|---|---|---|
| HAT (Chen et al., 2023) | 30.45/0.8448 | 31.43/0.8751 | 30.63/0.8725 | 29.99/0.8596 | 31.86/0.9146 |
| HAT+LDT | 31.34/0.8630 | 31.39/0.8771 | 31.47/0.8896 | 30.29/0.8713 | 32.32/0.9208 |
| SAFMN (Sun et al., 2023) | 30.46/0.8449 | 31.44/0.8753 | 30.64/0.8729 | 29.99/0.8597 | 31.86/0.9148 |
| SAFMN+LDT | 31.03/0.8583 | 30.92/0.8712 | 31.06/0.8835 | 29.99/0.8727 | 31.88/0.9209 |
| MambaIR (Guo et al., 2024a) | 30.81/0.8688 | 30.81/0.8850 | 30.22/0.8753 | 30.01/0.8737 | 30.93/0.8617 |
| MambaIR +LDT | 31.36/0.8611 | 32.15/0.8880 | 31.51/0.8865 | 30.57/0.8705 | 32.80/0.9246 |

## D.5 PERFORMANCE COMPARISON ON DIFFERENT DATA DISTRIBUTIONS FOR IMAGE SUPER-RESOLUTION TASK

As shown in Table 12, to validate the robustness of the LDT method against varying data distributions in the super-resolution task, we trained the MambaIR network using the Pan data branch as the source domain and evaluated it on B100, Manga109, Set5, Set14, and Urban100. The results demonstrate the strong generalization capability of the LDT method across different data distributions.

Table 12: Performance Comparison on Different Data Distributions for Image Super-Resolution

| Method | B100 | Manga109 | Set5 | Set14 | Urban100 |
|---|---|---|---|---|---|
| LPFT | 24.13/0.6551 | 23.79/0.8135 | 25.19/0.7713 | 23.58/0.6735 | 21.77/0.6717 |
| DeFT | 25.14/0.6885 | 24.53/0.8306 | 26.29/0.8001 | 24.94/0.7157 | 22.91/0.7113 |
| LDT | 25.38/0.6938 | 24.70/0.8467 | 26.44/0.8100 | 25.03/0.7214 | 23.03/0.7261 |

## D.6 PERFORMANCE COMPARISON ON DIFFERENT DATA DISTRIBUTIONS FOR NLP TASK

As shown in Table 13, to verify LDT's effectiveness in NLP tasks, we utilized the Amazon review dataset specifically designed for NLP domain generalization research. It contains four data branches: DVD, Kitchen, Electronics, and Books. We used the first three as source domains and the last as the target domain. LDT demonstrates clear performance improvements compared to the DeFT method.

## D.7 Performance comparison on different data distributions for image classification task

To verify the effectiveness of LDT on high-level vision tasks, we evaluate LDT's performance on image classification. As shown in Table.14, LDT demonstrates consistent performance improvements across different data branches. The most significant improvement occurs in the V branch, where LDT achieves a 5.19% accuracy gain over baseline methods when using ResNet-18 as the backbone network.

## D.8 Performance comparison on different data distributions for semantic segmentation task

As shown in Table.15, we evaluate the performance of the LDT method on semantic segmentation tasks. LDT demonstrates consistent performance improvements across different data branches.

### D.8.1 Ablation experiments on unstable layer partitioning ratios

To investigate how the unstable layer partitioning ratio $Ratio^U$ affects network generalization, we sample values between 0.1 and 0.9 at intervals of 0.2 (with finer 0.1 intervals near the optimal ratio), using these sampled values as the partitioning thresholds. As shown in Tables 16 and 17, the network achieves optimal generalization performance at selection ratios of 0.4 or 0.5 for image super-resolution, while a ratio of 0.7 yields the best performance for the semantic segmentation task.

Table 13: DG for NLP tasks.

| Method | Source Domain | Target Domain: Book |
|---|---|---|
| FT | | 0.8725 |
| DeFT | DVD, Kitchen, Electronics | 0.8800 |
| LDT | | 0.9000 |

Table 14: DG for image classification.

| Network | C | L | V | S | Mean |
|---|---|---|---|---|---|
| Resnet18 + FT | 0.9929 | 0.7363 | 0.6341 | 0.7941 | 0.7894 |
| Resnet18 + DeFT (Pahk et al., 2025) | 0.9965 | 0.7439 | 0.6433 | 0.8133 | 0.7992 |
| Resnet18 + LDT | 0.9965 | 0.7514 | 0.6860 | 0.8281 | 0.8155 |
| Resnet50 +FT | 0.9929 | 0.7345 | 0.6371 | 0.8148 | 0.7949 |
| Resnet50 + DeFT (Pahk et al., 2025) | 0.9929 | 0.7345 | 0.6418 | 0.8222 | 0.7978 |
| Resnet50 + LDT | 1.0000 | 0.7684 | 0.7058 | 0.8415 | 0.8289 |
| Vit +FT | 0.9965 | 0.7797 | 0.6570 | 0.8207 | 0.8135 |
| Vit + DeFT (Pahk et al., 2025) | 0.9929 | 0.7589 | 0.6433 | 0.7807 | 0.7940 |
| Vit + LDT | 1.0000 | 0.7589 | 0.6951 | 0.8400 | 0.8235 |
| Vision Mamba + FT | 1.0000 | 0.7684 | 0.6600 | 0.7956 | 0.8060 |
| Vision Mamba + DeFT (Pahk et al., 2025) | 1.0000 | 0.7759 | 0.6768 | 0.8430 | 0.8239 |
| Vision Mamba + LDT | 1.0000 | 0.7928 | 0.7043 | 0.8326 | 0.8324 |

Table 15: DG for semantic segmentation.

| Method | Source domain | Target domain 1 | Target domain 2 |
|---|---|---|---|
| FT | | BDD100K:30.7881 | Mapillary:34.9942 |
| DeFT (Pahk et al., 2025) | Cityscapes | BDD100K:42.4037 | Mapillary:48.3825 |
| LDT | | BDD100K:43.6769 | Mapillary:51.6588 |

Table 16: Ablation experiments on stable/unstable layer partitioning criteria for Super-resolution task

| Ratio | Pan | Sony | DSC | IMG | Canon |
|---|---|---|---|---|---|
| 0.1 | 30.73/0.8506 | 31.79/0.8828 | 30.89/0.8775 | 30.19/0.8633 | 32.10 /0.9184 |
| 0.3 | 31.28/0.8589 | 31.86/0.8800 | 31.41/0.8854 | 30.36/0.8681 | 32.44/0.9191 |
| 0.4 | 31.36/0.8611 | **32.15/0.8880** | **31.51**/0.8865 | **30.57**/0.8705 | **32.80/0.9246** |
| 0.5 | **31.37/0.8633** | 31.69/0.8805 | 31.43/**0.8883** | 30.36/0.8724 | 32.54/0.9237 |
| 0.6 | 31.25/0.8632 | 31.32/0.8755 | 31.24/0.8877 | 30.21/0.8731 | 32.40/0.9229 |
| 0.7 | 31.28/0.8628 | 31.24/0.8741 | 31.26/0.8875 | 30.20/**0.8735** | 32.44/0.9237 |
| 0.9 | 31.04/0.8623 | 31.00/0.8712 | 31.04/0.8858 | 29.95/0.8723 | 32.07/0.9209 |

Table 17: Ablation Study on the Impact of Layer Partitioning Ratios for Semantic Segmentation task

| Ratio | Target Domain 1 (mIOU) | Target Domain 2 (mIOU) |
|---|---|---|
| 0.1 | 38.155 | 43.852 |
| 0.2 | 38.452 | 42.797 |
| 0.3 | 42.587 | 50.547 |
| 0.4 | 44.851 | 53.075 |
| 0.5 | 44.766 | 53.647 |
| 0.6 | 43.284 | 53.687 |
| 0.7 | 45.420 | 55.733 |
| 0.8 | 44.445 | 54.448 |
| 0.9 | 45.386 | 55.444 |

# E    PSEUDOCODE OF OVERALL TRAINING PIPLINE

We formalize LDT's complete training procedure through pseudocode: Algorithm.1 presents the identification of stable and unstable layers, while Algorithm 2 details the gradient isolation and stabilization mechanism. Algorithm.3 presents the overall training procedure of LDT-S.

**Input:** The source domain subset $D_1^{So}$ contains input samples $x$ and ground-truth labels $\overline{y}$, network $M$, ratio of unstable layers $Ratio^U$;
**Output:** Initialized network $M$, name set of stable layers $Name^S$, name set of unstable layers $Name^U$;

```
// Warm-up stage
```
1   $\tilde{M} = Frozen\_by\_name(M, Backbone\_name)$ // Freeze the backbone network
2   **for** $i \leftarrow 1$ **to** $N_{warm-up}$ **do**
3      $y = \tilde{M}(x)$;
4      $Loss = func(y, \overline{y})$; // $func$ is L1 loss for SR tasks, cross-entropy or L2 loss for high-level tasks
5      $Loss.backward()$; // Update prediction head
6   **end**
```
// Layer selection stage
```
7   $M = Unfreeze(\tilde{M})$;
8   $Grad = Collect\_network\_gradients(M)$; // Collect gradients over all samples.
9   **for** $i \leftarrow 1$ **to** $len(M)$ **do**
     // Compute the gradient variance for each layer.
10     $Var_i = Comput\_var(Grad)$;
11   **end**
12   $Name^U = Top\_N(Var, Ratio^U, M)$;
13   $Name^S = Name^{All} - Name^U$; // $Name^{All}$: name set of all layers
14   **return** $M, Name^S, Name^U$;

**Algorithm 1:** Identification of stable and unstable layers

---

**Input:** Another source domain subset $D_2^{So}$ contains input samples $x$ and ground-truth labels $\overline{y}$, network $M$, name set of stable layers $Name^S$, name set of unstable layers $Name^U$;
**Output:** Trained network $MC$;
1   **for** $i \leftarrow 1$ **to** $N_{training}$ **do**
2      $PM, AM = Copy(M)$;
3      $\tilde{PM} = Frozen\_by\_name(PM, Name^U)$; // where $\tilde{PM} = \{PL^S, \tilde{PL}^U\}$
4      $\tilde{AM} = Frozen\_by\_name(AM, Name^S)$; // where $\tilde{AM} = \{\tilde{AL}^S, AL^U\}$
5      $y^P = \tilde{PM}(x)$;
6      $y^A = \tilde{AM}(x)$;
7      $Loss^P = func(y^P, \overline{y})$;
8      $Loss^A = func(y^A, \overline{y})$;
9      $Loss^P.backward()$; // Update unfrozen stable layers $PL^S$ in the primary network.
10     $Loss^A.backward()$; // Update unfrozen unstable layers $AL^U$ in the auxiliary network.
     // DPU
11     $W^S, W^U = Get\_Update\_Coefficients(Var^S, Var^U)$; // Refer to Eq.9 and 10.
12     $Update\_EMA(\tilde{PL}^U, AL^U, W^U)$; // Update frozen unstable layers $\tilde{PL}^U$ in the primary network. Refer to Eq.11.
13     $Update\_EMA(\tilde{AL}^S, PL^S, W^S)$; // Update frozen stable layers $\tilde{AL}^S$ in the auxiliary network.
14   **end**
15   $MC = Cat\{\tilde{AL}^S, \tilde{PL}^U\}$; // Concatenate frozen stable layers from auxiliary network and unstable layers from primary network.
16   **return** $MC$;

**Algorithm 2:** Gradient isolation and stabilization

**Input:** Another source domain subset $D_2^S$ contains input samples $x$ and ground-truth labels $\overline{y}$, network $M$, name set of stable layers $Name^S$, name set of unstable layers $Name^U$, $ss$ side-switching interval;

**Output:** Weight buffer $WB$;

```
// where M = {L^S, L^U}, WB = {WBL^S, WBL^U}
```

1  $WB = Copy(M)$ `// Weight buffer`
2  $Flag = Stable$;
3  $\tilde{M} = Frozen\_by\_name(M, Name^S)$;
4  $W^S, W^U = Get\_Update\_Coefficients(Var^S, Var^U)$; `// Refer to Eq.9 and 10.`
5  **for** $i \leftarrow 1$ **to** $N_{training}$ **do**
6     $y = \tilde{M}(x)$;
7     $Loss = func(y, \overline{y})$;
8     $Loss.backward()$; `// Update unfrozen layers.`
9     **if** $Flag == Stable$ **then**
10        $Update\_EMA(WBL^U, L^U, W^U)$; `// Update weight buffer.`
11     **end**
12     **else**
13        $Update\_EMA(WBL^S, L^S, W^S)$;
14     **end**
     `// Switch the training sides.`
15     **if** $i\%ss == 0$ **then**
16        $Loda\_weight(M, WB)$; `// Load the weights from EM into M.`
17        **if** $Flag == Stable$ **then**
18           $\tilde{M} = Unfrozen\_by\_name(M, Name^S)$;
19           $Flag = Unstable$;
20           $\tilde{M} = Frozen\_by\_name(M, Name^U)$;
21        **end**
22        **else**
23           $\tilde{M} = Unfrozen\_by\_name(M, Name^U)$;
24           $Flag = Stable$;
25           $\tilde{M} = Frozen\_by\_name(M, Name^S)$;
26        **end**
27     **end**
28 **end**
29 **return** $WB$;

**Algorithm 3:** Gradient isolation and stabilization with a single active network

# F VISUALIZATION RESULTS

As shown in Figure.8, we compared the proposed LDT with other domain generalization methods. LDT demonstrated superior detail restoration and noise suppression.

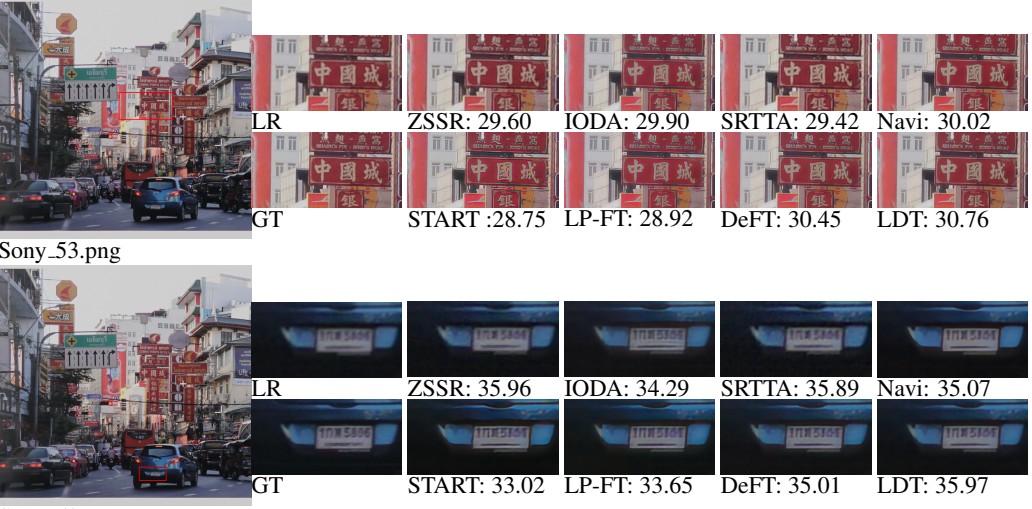

Figure 8: Visual comparison. The large image on the left is the LR image, and the sub-images on the right are LR, ZSSR (Shocher et al., 2018), IODA (Tang & Yang, 2024), SRTTA (Deng et al., 2023), Wang et al. (2024a)(Navi)(first row), GT, START (Guo et al., 2024b), LP-FT (Kumar et al., 2022), DeFT (Pahk et al., 2025), LDT (second row). The value following the name represents the PSNR metric of the current patch. Please zoom-in on screen.

## G    STATEMENT ON LLM USAGE

We used a Large Language Model (LLM), specifically ChatGPT, solely for language polishing and improving the readability of the manuscript. The LLM was not used to generate ideas, conduct experiments, analyze results, or contribute to the research methodology. All scientific content, including the conceptualization, design, implementation, and validation of the work, was entirely carried out by the authors.

