# OpenReview forum: "LDT: Layer-Decomposition Training Makes Networks More Generalizable"
_ICLR.cc/2026/Conference — ICLR 2026 Poster_

### Official Review · Reviewer_kBcr · 2025-10-31

**Soundness:** 3
**Presentation:** 3
**Contribution:** 3
**Rating:** 6
**Confidence:** 3

**Summary:**

The paper proposes Layer-Decomposition Training (LDT), a fine-grained, layer-wise method to improve domain generalization by explicitly separating “stable” and “unstable” layers based on gradient variance measured after a warm-up. Training uses two copies of the model with cross-freezing (stable layers train in the primary network; unstable layers train in the auxiliary network), while frozen counterparts are updated via EMA. A Dynamic Parameter Update (DPU) strategy adapts EMA coefficients per layer according to each layer’s relative variance rank. Experiments span super-resolution (DRealSR across cameras), classification (VLCS), and semantic segmentation (Cityscapes/BDD/Mapillary), and multiple backbones (CNN/Transformer/Mamba). Reported gains are strongest on SR, with modest training-time overhead and identical inference memory to baseline.

**Strengths:**

1. **Clear problem framing**: The paper targets a concrete weakness of coarse parameter partitioning (backbone vs. head) in prior work (LP-FT, DeFT) and motivates finer granularity using measured layer-wise gradient variance. The dual-branch cross-freezing setup is sensible and well-aligned with the goal of mitigating unstable-to-stable gradient interference.
2. **Method simplicity & generality**: LDT and DPU are architecture-agnostic and slot into standard fine-tuning loops with minimal code complexity (once gradients are collected). Conceptually straightforward; likely portable to many vision backbones.
3. **Empirical signal on SR**: On DRealSR, LDT and especially LDT+DPU improve PSNR across several target domains; the Canon branch shows a large jump ($\approx$ +1.9 dB over baseline). The ablations comparing variance vs. mean vs. random partitioning support the central design choice.

**Weaknesses:**

1. Metric inconsistencies: On SR, SSIM sometimes decreases even when PSNR increases (e.g., Pan and IMG branches in Table 1). The paper does not analyze why nor whether perceptual quality suffers. This matters for claims of “superior generalization capability.”

2. Hyperparameter sensitivity: The ratio of unstable layers (Ratio_U) is central but lacks guidance: how is it chosen, how sensitive are results, and does it vary by architecture/task? The normalized-variance vs variance choice is argued for simplicity, yet Table 2 shows Var/Mean sometimes wins; the trade-off merits deeper discussion.

3. Writing/Presentation issues：Several language glitches (e.g., “pm” instead of “±”), accidental filename text (“panasonic 132.png”), and inconsistent naming (“Layer-Decomposition” vs. “Layer-decoupled” in Appendix) distract from the contribution and could impede reproducibility.

4. Novelty & Relation to Prior Work: Relative to LP-FT/DeFT, the novelty is finer-grained, data-driven layer partitioning and ranked EMA coefficients. This is incremental but non-trivial: the idea that some backbone layers can be less stable than head layers is interesting and empirically supported. Comparisons to other stability-oriented training schemes (e.g., sharpness- or noise-based regularizers) would situate the contribution better.

**Questions:**

1. How exactly is gradient variance computed per layer? Is it over per-parameter gradients aggregated by mean/std? Over mini-batches or single examples? Any normalization by parameter scale?

2. DPU coefficients. Did any layers receive W=1? If so, are those layers effectively frozen? Could you share the distribution of learned W across layers and runs?

---

> ### Author Response · Authors · 2025-11-20
> **Official Comment by Authors [1/2]**
>
> We thank you for your detailed feedback. We will address all of your comments and questions point by point below. Additionally, we have revised the manuscript accordingly. Please download the updated version for your reference.
>
> **1.Metric inconsistencies: On SR, SSIM sometimes decreases even when PSNR increases (e.g., Pan and IMG branches in Table 1). The paper does not analyze why nor whether perceptual quality suffers. This matters for claims of “superior generalization capability.”**
>
> (1) Taking datasets A and B as an example: domain generalization methods are trained on dataset A, and the trained model is then evaluated on dataset B. For the super-resolution task, we select the optimal model weights based on their PSNR performance on dataset A during training, and then use these selected weights for inference on dataset B. Since we rely solely on the PSNR metric for weight selection and do not optimize for SSIM, the chosen weights may lead to less significant improvement in SSIM on some data branches.
>
> (2) We have validated the effectiveness of our LDT method across various task scenarios (image super-resolution, classification, and semantic segmentation), different network architectures (ResNet18, ResNet50, ViT, Mamba), and diverse data distributions (including DRealSR, VLCS, BDD100K, Cityscapes, and Mapillary datasets). LDT consistently demonstrates stable performance improvements across all these settings.
>
>
> **2.Hyperparameter sensitivity: The ratio of unstable layers (Ratio_U) is central but lacks guidance: how is it chosen, how sensitive are results, and does it vary by architecture/task? The normalized-variance vs variance choice is argued for simplicity, yet Table 2 shows Var/Mean sometimes wins; the trade-off merits deeper discussion.**
>
> (1) As shown in Table 1 of this rebuttal, we have supplemented experiments analyzing the impact of different layer partitioning ratios on network performance in semantic segmentation tasks. The results indicate that the network achieves optimal performance when the ratio is set to 0.7.
>
> (2) Regarding the selection method for the stable/unstable layer partitioning ratio, we first execute the first stage of the LDT method: we compute and rank the gradient variances of each network layer across 300 samples, then visualize them to identify the variance jump point (which requires meeting two conditions: rapid variance increase and location near the middle of the overall ranking). In SR tasks using Mamba networks, the variance jump point is near 0.4, while in semantic segmentation tasks using ResNet networks, the variance jump point is near 0.7.
>
> (3) Unstable parameters are highly sensitive to data distribution shifts. Slight changes can cause significant fluctuations in these parameters, resulting in large gradient means. This leads to low Var/Mean values, causing these layers to be misclassified as stable and ultimately reducing the network's generalization performance. Therefore, variance remains the optimal criterion for partitioning.
>
> >Table 1. Ablation Study on the Impact of Layer Partitioning Ratios for Semantic Segmentation. This study uses Cityscapes as the source domain, with BDD100K and Mapillary serving as Target Domain 1 and Target Domain 2, respectively.
> >|Ratio|Target domain 1 (mIOU)|Target domain 2 (mIOU)|
> >|-|-|-|
> >|0.1|38.155|43.852|
> >|0.2|38.452|42.797|
> >|0.3|42.587|50.547|
> >|0.4|44.851|53.075|
> >|0.5|44.766|53.647|
> >|0.6|43.284|53.687|
> >|0.7|45.420|55.733|
> >|0.8|44.445|54.448|
> >|0.9|45.386|55.444|
>
> **3.Writing/Presentation issues：Several language glitches (e.g., “pm” instead of “±”), accidental filename text (“panasonic 132.png”), and inconsistent naming (“Layer-Decomposition” vs. “Layer-decoupled” in Appendix) distract from the contribution and could impede reproducibility.**
>
> We thank you for pointing this out. We have revised the manuscript accordingly. Regarding the filename text, this was not an error but rather an intentional reference to facilitate readers in locating the specific comparison images within the dataset.

---

> > ### Author Response · Authors · 2025-11-20
> > **Official Comment by Authors [2/2]**
> >
> > **4.Novelty & Relation to Prior Work: Relative to LP-FT/DeFT, the novelty is finer-grained, data-driven layer partitioning and ranked EMA coefficients. This is incremental but non-trivial: the idea that some backbone layers can be less stable than head layers is interesting and empirically supported. Comparisons to other stability-oriented training schemes (e.g., sharpness- or noise-based regularizers) would situate the contribution better.**
> >
> > (1) We sincerely thank the reviewer for recognizing the novelty of our LDT method. Given the time constraints and the vast number of publications in domain generalization research, we found it challenging to identify all relevant comparative works. We would greatly appreciate if the reviewer could specify particular references to facilitate a more targeted comparison. However, due to the limited duration of the author-reviewer discussion window and the need to address comments from multiple reviewers, we will note these references and make every effort to complete the suggested experiments. If unable to finalize them before the window closes, we will include this analysis in an extended version of this work.
> >
> > (2) Notably, inspired by Reviewer e1yc's suggestion, we have developed a single-branch variant of our LDT method (LDT-S). LDT-S achieves lower computational cost by alternately freezing parts of the single network across timesteps, maintaining the requisite isolation between stable and unstable layers while ensuring the stabilization of the unstable ones. This approach significantly improves computational efficiency, substantially reducing both training time and memory consumption, while achieving performance competitive with the dual-branch DeFT method. (For detailed information about LDT-S, please refer to Appendix C in the revised manuscript)
> >
> >
> > **5.How exactly is gradient variance computed per layer? Is it over per-parameter gradients aggregated by mean/std? Over mini-batches or single examples? Any normalization by parameter scale?**
> >
> > We feed 300 samples into the network and compute the gradients for each layer across these 300 samples. We then calculate the mean and variance of the gradients for each layer based on these 300 gradient instances. Throughout this gradient acquisition process, no parameter updates are performed.
> >
> > **6.DPU coefficients. Did any layers receive W=1? If so, are those layers effectively frozen? Could you share the distribution of learned W across layers and runs?**
> >
> > (1) We imposed constraints on W to ensure that no layer receives exactly W=1, though some layers may receive values like 0.999..., which are extremely close to 1. If W were exactly 1, the parameters of that layer would remain frozen during training.
> >
> > (2) We have included the distribution of W values in the revised manuscript. Please download the updated version and refer to Appendix D.2 for detailed visualization.

---

> > > ### Comment · Reviewer_kBcr · 2025-11-25
> > > **Response to the authors**
> > >
> > > I appreciate the authors for clarification of my question. Most of my questions are addressed and I decided to maintain my score.

---

> > > > ### Author Response · Authors · 2025-11-26
> > > >
> > > > We appreciate your quick response and positive recognition of our work.

---

### Official Review · Reviewer_cWBr · 2025-11-01

**Soundness:** 3
**Presentation:** 3
**Contribution:** 3
**Rating:** 6
**Confidence:** 4

**Summary:**

This paper proposes a new training strategy, LDT (Layer-Decomposition Training), to improve domain generalization performance. It critiques existing methods for their coarse-grained partitioning of networks into 'stable' backbones and 'unstable' heads. LDT, in contrast, identifies stable/unstable layers at a fine-grained, individual layer level based on gradient variance. LDT uses a dual-branch (primary/auxiliary) network structure with a cross-freezing mechanism to isolate gradient interference from unstable layers, preventing it from disrupting the updates of stable layers. Furthermore, it introduces the DPU (Dynamic Parameter Update) strategy. Instead of using a fixed coefficient for parameter updates (EMA), DPU dynamically adjusts the update coefficient for each layer based on its variance level. This allows low-variance (stable) layers to learn more efficiently and high-variance (unstable) layers to be stabilized more effectively. The authors demonstrate that LDT achieves superior generalization performance compared to existing methods across diverse vision tasks (e.g., Super-Resolution, Classification, Semantic Segmentation) and various architectures (CNN, Transformer, Mamba).

**Strengths:**

1. Intuitive Problem Definition: The paper clearly points out the problem with existing methods (like DeFT) that simply separate the 'backbone' and 'head'. Fig 2(a) visually demonstrates that some backbone layers are even more unstable than the head, strongly justifying the core idea that a fine-grained, 'layer-wise' separation based on 'gradient variance' is necessary.
2. Excellent Versatility (Task- & Architecture-Agnostic): The proposed methodology was successfully applied not only to low-level vision tasks (Super-Resolution) but also to high-level vision tasks (Classification, Segmentation). Furthermore, it demonstrated consistent performance improvements across diverse models, including CNN, Transformer, and the recent Mamba architecture, proving that LDT is a robust generalization 'strategy' not limited to specific conditions.
3. Robust Experimental Validation: The ablation studies (Tables 1 and 2) are very well-designed, clearly isolating and verifying the effectiveness of LDT's core component (gradient variance-based partitioning) and the additional contribution of DPU (LDT+DPU > LDT > Baseline). This demonstrates that each proposed component makes a tangible and significant contribution.

**Weaknesses:**

1. Increased Training Cost: LDT uses a dual-branch architecture, which significantly increases training memory (approx. 1.3x based on Table 3) and training time (approx. 1.8x) compared to the baseline. This could be a practical barrier for training large-scale models.
2. Sensitivity to Key Hyperparameter: The method introduces a new key hyperparameter, the 'unstable layer ratio ($Ratio^U$)'. According to Table 9 in the appendix, performance is sensitive to this ratio (optimal around 0.4-0.5), requiring careful tuning when applying it to new tasks or datasets.
3. Fixedness of Layer Selection: The distinction between stable and unstable layers is determined only once—immediately after the warm-up stage—and remains fixed throughout the entire training process. However, as training progresses, the stability of certain layers may change dynamically.

**Questions:**

1. The DPU update coefficients (Eq. 10) are determined by a linear mapping based on 'rank' rather than the variance values themselves. This method (e.g., 0.99 + 0.01 * Rank) appears somewhat empirical. I am curious if you considered other functions that directly utilize the actual variance values (e.g. proportional to the normalized variance), and if so, why the current rank-based approach was found to be optimal.
2. The increased training cost from the dual-branch setup could be prohibitive for larger models. Are there potential ways to reduce this cost while retaining the benefits of LDT, such as parameter sharing between the two networks or knowledge distillation?
3. The stable/unstable status of the layers is fixed throughout the entire training process. If a dynamic LDT were applied, for instance, by re-measuring the gradient variance and updating the partitions periodically, could this lead to further performance improvements? Or, conversely, would it risk harming training stability?

---

> ### Author Response · Authors · 2025-11-20
> **Official Comment by Authors [1/2]**
>
> We thank you for your detailed feedback. We will address all of your comments and questions point by point below. Additionally, we have revised the manuscript accordingly. Please download the updated version for your reference.
>
> **1.Sensitivity to Key Hyperparameter: The method introduces a new key hyperparameter, the 'unstable layer ratio (Ratio^{u})'. According to Table 9 in the appendix, performance is sensitive to this ratio (optimal around 0.4-0.5), requiring careful tuning when applying it to new tasks or datasets.**
>
> Regarding the selection method for the stable/unstable layer partitioning ratio, we first execute the first stage of the LDT method: we compute and rank the gradient variances of each network layer across 300 samples, then visualize them to identify the variance jump point (which requires meeting two conditions: rapid variance increase and location near the middle of the overall ranking). In SR tasks using Mamba networks, the variance jump point is near 0.4, while in semantic segmentation tasks using ResNet networks, the variance jump point is near 0.7.
>
> **2.The DPU update coefficients (Eq. 10) are determined by a linear mapping based on 'rank' rather than the variance values themselves. This method (e.g., 0.99 + 0.01 * Rank) appears somewhat empirical. I am curious if you considered other functions that directly utilize the actual variance values (e.g. proportional to the normalized variance), and if so, why the current rank-based approach was found to be optimal.**
>
> This is an excellent observation. We fully agree with the reviewer's perspective that relying solely on ranking to map gradients to coefficients inevitably results in information loss, as it reduces the complex distribution of layer-wise variances to a linear projection. However, implementing a continuous mapping from variance to weight coefficients requires careful design. As shown in Table 1 of this rebuttal, our preliminary attempt of using normalized variance directly as the weighting coefficient led to performance degradation. We strongly agree with the reviewer's viewpoint and find this direction highly interesting; it will be included as an important extension of our future work.
>
> >Table 1. Ablation Study on Different Weight Coefficient W Projection Methods
> >|Method|Pan|Sony|DSC|IMG|Canon|
> >|-|-|-|-|-|-|
> >|Baseline| 30.81/0.8688|30.81/0.8850|30.22/0.8753|30.01/0.8737|30.93/0.8617|
> >|Normalized variance|31.09/0.8624|31.01/0.8702|30.05/0.8726| 31.07/0.8855|32.31/0.9227|
> >|Ours|31.36/0.8611|32.15/0.8880|31.51/0.8865|30.57/0.8705|32.80/0.9246|
>
>
> **3.The increased training cost from the dual-branch setup could be prohibitive for larger models. Are there potential ways to reduce this cost while retaining the benefits of LDT, such as parameter sharing between the two networks or knowledge distillation?**
>
> (1) Inspired by the reviewer's suggestion (e1yc), we have developed an improved version of LDT called the single-branch LDT method (LDT-S). LDT-S achieves lower computational cost by alternately freezing parts of the single network across timesteps, maintaining the requisite isolation between stable and unstable layers while ensuring the stabilization of the unstable ones. (For detailed information about LDT-S, please refer to Appendix C in the revised manuscript)
>
> (2) As shown in Table 2 of this rebuttal, LDT-S effectively improves computational efficiency, significantly reducing both training time and memory usage. Moreover, Table 3 of this rebuttal demonstrates that LDT-S achieves performance comparable to the dual-branch DeFT method while offering superior computational efficiency.
>
> (3) Finally, we find your proposed solution equally interesting and potentially more effective. The knowledge distillation approach might even reduce computational memory consumption below single-network levels. We are considering this as a promising direction for future research and thank you sincerely for the valuable suggestion.
>
> >Table 2 Ablation experiments on training/inference efficiency. The task is image super-resolution,
> with training and inference patch sizes set to 48×48 and 200×200 pixels respectively. The network
> architecture is based on MambaIR
> >|Method|Training memory (GB)|Inf memory (GB)|Training time (s)|Inf time (s)|
> >|-|-|-|-|-|
> >|Baseline|15.27|2.7|0.6912|658.3287|
> >|DeFT|20.30|2.7|1.2566|653.9900|
> >|LDT-S|15.21|2.7|0.6920|649.5462|
> >|LDT|20.25|2.7|1.2608|643.6736|
>
> >Table 3 Performance comparison
> >|Method|Pan|Sony|DSC|IMG|Canon|
> >|-|-|-|-|-|-|
> >|Baseline|30.81/0.8688|30.81/0.8850|30.22/0.8753| 30.01/0.8737|30.93/0.8617|
> >|DeFT|31.27/0.8632|31.61/0.8801|31.34/0.8875| 30.31/0.8726|32.40/0.9247|
> >|LDT-S|31.2982/0.8600|31.4621/0.8793| 31.3679/0.8878|30.3735/0.8729| 32.4920/0.9228|
> >|LDT|31.36/0.8611|32.15/0.8880|31.51/0.8865|30.57/0.8705| 32.80/0.9246|

---

> > ### Author Response · Authors · 2025-11-20
> > **Official Comment by Authors [2/2]**
> >
> > **4.The stable/unstable status of the layers is fixed throughout the entire training process. If a dynamic LDT were applied, for instance, by re-measuring the gradient variance and updating the partitions periodically, could this lead to further performance improvements? Or, conversely, would it risk harming training stability?**
> >
> > This is a truly interesting idea! We sincerely thank the reviewer for this valuable suggestion. As shown in Table 4 of this rebuttal, we have supplemented our ablation studies with experiments on dynamic partitioning of stable/unstable layers, where the LDT method reassigns these layers every 5,000 training steps. The dynamic partitioning strategy achieves higher PSNR and SSIM scores than the fixed partitioning approach across the Pan, DSC, and Canon data branches. This method better adapts to the training process by dynamically identifying stable and unstable layers according to the evolving parameter states, demonstrating improved adaptability.
> >
> > >Table 4. Ablation experiments on layer partitioning strategies
> > >|Method|Pan|Sony|DSC|IMG|Canon|
> > >|-|-|-|-|-|-|
> > >|Baseline|30.81/0.8688|30.81/0.8850|30.22/0.8753|30.01/0.8737|30.93/0.8617|
> > >|Static|31.36/0.8611|32.15/0.8880|31.51/0.8865|30.57/0.8705|32.80/0.9246|
> > >|Dynamic|31.4123/0.8636|31.8528/0.8826|31.5299/0.8892|30.4965/0.8729|32.7655/0.9250|

---

### Official Review · Reviewer_e1yc · 2025-11-01

**Soundness:** 2
**Presentation:** 2
**Contribution:** 3
**Rating:** 4
**Confidence:** 3

**Summary:**

This paper presents Layer Decomposition Training (LDT), a new domain generalization framework that improves network robustness by performing fine grained layer- wise separation of stable and unstable parameters. Existing works such as LP-FT and DeFT only distinguish between backbone and prediction head, which the authors argue is too coarse and leads to misclassification of unstable parameters.
 LDT first identifies unstable layers by measuring gradient variance across training samples and then adopts a dual branch cross freezing mechanism to isolate gradients between stable and unstable layers. To further enhance adaptability, a Dynamic Parameter Update (DPU) strategy is introduced, which adaptively adjusts EMA coefficients per layer based on gradient fluctuation magnitude.
 Extensive experiments are conducted across multiple domains including super resolution, classification, and semantic segmentation and across architectures such as CNNs, Transformers, and Mamba models. Results show consistent generalization improvements over state-of-the-art methods like DeFT and START.

**Strengths:**

The idea of layer wise decomposition of parameters guided by gradient variance is both conceptually novel and practically meaningful. Prior works (LP-FT, DeFT) only perform module level partitioning, whereas this work introduces a more granular and data driven approach. The Dynamic Parameter Update (DPU) is an innovative extension to EMA based stabilization, turning the update coefficient into a layer adaptive parameter, which enhances stability and learning efficiency. Theoretical motivation is well supported: the authors provide a gradient correlation analysis between stable and unstable layers and formal proofs (Appendix B) showing how unstable gradients propagate perturbations.

**Weaknesses:**

Although the paper provides a theoretical derivation of gradient interference, the proof is limited to a simplified two module case (stable vs. unstable). Extending this to multi layer interactions or real architectures would make the analysis more convincing. The dual network training doubles the model instances during training, increasing memory from 15.27 GB to 20.25 GB (Table 3). Although inference cost is unaffected, a deeper analysis on scalability to larger networks (e.g., ViT-L or Swin-L) is missing.

**Questions:**

DPU assigns update coefficients based on sorted variance rankings. Would a continuous mapping (e.g., normalization based scaling instead of discrete ranking) further improve stability? and could the authors explore a single network variant that integrates DPU like adaptation without duplicating the model? This might broaden applicability to resource constrained environments.

In today's era of large models, can this method be applied to large model architectures?

---

> ### Author Response · Authors · 2025-11-20
> **Official Comment by Authors [1/2]**
>
> We thank you for your detailed feedback. We will address all of your comments and questions point by point below. Additionally, we have revised the manuscript accordingly. Please download the updated version for your reference.
>
> **1.Although the paper provides a theoretical derivation of gradient interference, the proof is limited to a simplified two module case (stable vs. unstable). Extending this to multi layer interactions or real architectures would make the analysis more convincing.**
>
> We have extended the theoretical derivation of gradient interference to the multi-layer case. To ensure readability, please download the revised manuscript and refer to the updated Appendix B.
>
> **2.The dual network training doubles the model instances during training, increasing memory from 15.27 GB to 20.25 GB (Table 3). Although inference cost is unaffected, a deeper analysis on scalability to larger networks (e.g., ViT-L or Swin-L) is missing.**
>
> (1) Since HAT directly applies Swin to SR tasks, (As shown in Table 1 of this rebuttal) we have included the HAT-L network in the training/inference efficiency comparison for a fair evaluation. For LDT-S, please refer to our response to your Question 4.
>
> (2) While the introduction of dual networks inevitably increases computational overhead, approximately half of the network layers remain frozen during training (without parameter gradient updates), which keeps the additional computational cost within a relatively manageable range.
>
> >Table 1. Ablation experiments on training/inference efficiency. The task is image super-resolution,
> with training and inference patch sizes set to 48×48 and 200×200 pixels respectively.
> >|Network| Training memory (GB) | Inf memory (GB) | Training time (s) | Inf time (s) |
> >|-|-|-|-|-|
> >|HAT|15.17|5.9|0.2305|982.1352|
> >|HAT-DeFT|25.54|5.9|0.4551|984.5441|
> >|HAT-LDT|25.55|5.9|0.4424|987.4716|
> >|HAT-LDT-S|15.26|5.9|0.2313|985.6354|
> >|MambaIR|15.27|2.7|0.6912|658.3287|
> >|MambaIR-DeFT|20.30|2.7|1.2566|653.9900 |
> >|MambaIR-LDT|20.25|2.7|1.2608|643.6736|
> >|MambaIR-LDT-S|15.21|2.7|0.6920|649.5462|
>
> **3.DPU assigns update coefficients based on sorted variance rankings. Would a continuous mapping (e.g., normalization based scaling instead of discrete ranking) further improve stability?**
>
> This is an excellent suggestion. We fully agree with the reviewer's perspective that relying solely on ranking to map gradients to weights inevitably results in information loss, as it essentially reduces the complex distribution of layer-wise variances to a linear projection. However, implementing a continuous mapping from variance to weight coefficients requires careful design. As shown in Table 2 of this rebuttal, our preliminary attempt using normalized variance directly as the weight coefficient led to performance degradation. We completely agree with the reviewer's viewpoint and find this direction highly interesting; it will be included as an important extension of our future work.
>
> Table 2. Ablation Study on Different Weight Coefficient W Projection Methods
> |Method|Pan|Sony|DSC|IMG|Canon|
> |-|-|-|-|-|-|
> |Baseline| 30.81/0.8688| 30.81/0.8850|30.22/0.8753|30.01/0.8737| 30.93/0.8617|
> |Normalized variance|31.09/0.8624|31.01/0.8702|30.05/0.8726| 31.07/0.8855| 32.31/0.9227|
> |Ours| 31.36/0.8611| 32.15/0.8880|31.51/0.8865|30.57/0.8705| 32.80/0.9246|

---

> ### Author Response · Authors · 2025-11-20
> **Official Comment by Authors [2/2]**
>
> **4.could the authors explore a single network variant that integrates DPU like adaptation without duplicating the model? This might broaden applicability to resource constrained environments.**
>
> (1) We are very grateful to the reviewer for this insightful suggestion, which has inspired a significant improvement to our method. Following your advice, we have developed an enhanced version of LDT: the single-branch LDT method (LDT-S). LDT-S achieves lower computational cost by alternately freezing parts of the single network across timesteps, maintaining the requisite isolation between stable and unstable layers while ensuring the stabilization of the unstable ones. (For detailed information about LDT-S, please refer to Appendix C in the revised manuscript)
>
> (2) As shown in Table 3 of this rebuttal, LDT-S effectively improves computational efficiency, significantly reducing both training time and memory usage. Moreover, Table 4 of this rebuttal demonstrates that LDT-S achieves performance comparable to the dual-branch DeFT method while offering superior computational efficiency.
>
> >Table 3 Ablation experiments on training/inference efficiency. The task is image super-resolution,
> with training and inference patch sizes set to 48×48 and 200×200 pixels respectively. The network
> architecture is based on MambaIR.
> >|Method|Training memory (GB)|Inf memory (GB)|Training time (s)|Inf time (s)|
> >|-|-|-|-|-|
> >|Baseline|15.27|2.7|0.6912|658.3287|
> >|DeFT|20.30|2.7|1.2566|653.9900|
> >|LDT-S|15.21|2.7|0.6920|649.5462|
> >|LDT|20.25|2.7|1.2608|643.6736|
>
> >Table 4 Performance comparison
> >|Method|Pan|Sony|DSC|IMG|Canon|
> >|-|-|-|-|-|-|
> >|Baseline|30.81/0.8688|30.81/0.8850|30.22/0.8753| 30.01/0.8737|30.93/0.8617|
> >|DeFT|31.27/0.8632|31.61/0.8801|31.34/0.8875| 30.31/0.8726|32.40/0.9247|
> >|LDT-S|31.30/0.8600|31.46/0.8793| 31.37/0.8878|30.37/0.8729| 32.49/0.9228|
> >|LDT|31.36/0.8611|32.15/0.8880|31.51/0.8865|30.57/0.8705| 32.80/0.9246|
>
> **5.In today's era of large models, can this method be applied to large model architectures?**
>
> Our method remains applicable in the context of large models. Fine-tuning is a common strategy for large models, and we can apply the LDT method specifically to the fine-tuned parameters. This can be achieved by duplicating the fine-tuned parameters into two branches and applying alternating freezing. Alternatively, in Mixture-of-Experts (MoE) based large models, LDT can be applied to individual experts. These approaches can effectively enhance the robustness of fine-tuned models with only a minimal increase in training parameters.
>
> Additionally, LDT-S can also be utilized for optimizing the training of large models. It offers high computational efficiency while maintaining strong robustness.

---

### Official Review · Reviewer_GLbA · 2025-11-01

**Soundness:** 2
**Presentation:** 1
**Contribution:** 1
**Rating:** 2
**Confidence:** 4

**Summary:**

This paper proposes Layer Decomposition Training (LDT) for domain generalization by detecting “unstable” layers identified via high gradient variance and separating their influence from “stable” layers. LDT trains two parallel copies of the network with cross-freezing and an EMA pathway to prevent interference, and introduces a Dynamic Parameter Update that adapts each frozen layer’s EMA coefficient according to its variance rank. The method is architecture and task agnostic, delivering gains on super-resolution, classification, and segmentation without any added inference overhead.

**Strengths:**

- The paper methodology of using the Top-n layers with highest gradient variance as the unstable layer is easy to follow and easy to implement in practice.

**Weaknesses:**

- The first concern is novelty. Dual-branch training already exists [1]; the difference here is a layer-wise variant that selects “unstable” layers via a top-n gradient-variance threshold, which feels ad-hoc.
- The experimental evidence is limited to medium-scale vision benchmarks, with no large-scale tasks (e.g., ImageNet classification), making it hard to assess scalability and practical impact.
- The method is presented as task-agnostic, yet there are no natural-language experiments (e.g., WikiText-103 language modeling), which would help demonstrate broader applicability.
- Reported gains over DeFT [1] are modest (e.g., Table 4), suggesting limited improvement relative to prior dual-branch approaches.
- Sections 2.2.3 and 2.4 are unnecessarily long relative to their content, which impairs readability. Greater emphasis should be placed on more baseline comparisons against other domain generalization methods.

References:

[1] Jaehyun Pahk, Donghyeon Kwon, Seong Joon Oh, and Suha Kwak. Decoupled finetuning for domain generalizable semantic segmentation. ICLR, 2025.

**Questions:**

- Can you extend the evaluation to larger-scale and non-vision settings (e.g., ImageNet classification and an NLP benchmark such as WikiText-103), and outline any adaptations needed to apply LDT in these regimes?
- Can you expand the comparisons with other domain generalization methods on datasets beyond DRealSR benchmark?

---

> ### Author Response · Authors · 2025-11-20
> **Official Comment by Authors [1/2]**
>
> We thank you for your detailed feedback. We will address all of your comments and questions point by point below. Additionally, we have revised the manuscript accordingly. Please download the updated version for your reference.
>
> **1.The first concern is novelty. Dual-branch training already exists [1]; the difference here is a layer-wise variant that selects “unstable” layers via a top-n gradient-variance threshold, which feels ad-hoc.**
>
> (1) The proposed LDT method differs from existing approaches like DeFT in several important aspects:
>
> * We provide a theoretical analysis of how unstable layers perturb stable layers during gradient propagation, extending from two-layer to multi-layer network structures (see revised manuscript Appendix B).
> * Through visual analysis of network instability, we demonstrate that DeFT's parameter partitioning operates at an inadequate granularity, leading to misclassification.
> * Based on this finding, we propose a finer-grained, variance-guided method for partitioning stable and unstable layers.
> * While DeFT applies uniform update strategies across all layers, our analysis reveals significant variance disparities within both stable and unstable layer groups, leading to underutilization of amplitude information. To address this, we introduce a dynamic parameter update strategy.
> * Importantly, inspired by Reviewer e1yc's suggestion, we have developed a single-branch variant of our method (LDT-S). LDT-S achieves lower computational cost by alternately freezing parts of the single network across timesteps, maintaining the requisite isolation between stable and unstable layers while ensuring the stabilization of the unstable ones. This approach significantly improves computational efficiency (Table 1 of this rebuttal), reducing both training time and memory consumption to single-branch levels while maintaining competitive performance compared to dual-branch methods like DeFT (Table 2 of this rebuttal). **(For detailed information about LDT-S, please refer to Appendix C in the revised manuscript)**.
>
> (2) Our method is not simply an added module, but is derived from theoretical reasoning and visual statistical analysis that identifies the limitations in existing methods' partitioning granularity and information utilization. We subsequently address these issues through layer-wise partitioning, dynamic parameter updates, and the newly introduced single-branch LDT-S variant.
>
> (3) We aimed to develop a simple, plug-and-play method. Simplicity does not imply low innovation; rather, it enhances practical applicability. All source code for LDT, including the single-branch LDT-S variant, will be released within one week of paper acceptance.
>
>
> >Table 1 Ablation experiments on training/inference efficiency. The task is image super-resolution,
> with training and inference patch sizes set to 48×48 and 200×200 pixels respectively. The network
> architecture is based on MambaIR.
> >|Method|Training memory (GB)|Inf memory (GB)|Training time (s)|Inf time (s)|
> >|-|-|-|-|-|
> >|Baseline|15.27|2.7|0.6912|658.3287|
> >|DeFT|20.30|2.7|1.2566|653.9900|
> >|LDT-S|15.21|2.7|0.6920|649.5462|
> >|LDT|20.25|2.7|1.2608|643.6736|
>
>
> >Table 2 Performance comparison
> >|Method|Pan|Sony|DSC|IMG|Canon|
> >|-|-|-|-|-|-|
> >|Baseline|30.81/0.8688|30.81/0.8850|30.22/0.8753| 30.01/0.8737|30.93/0.8617|
> >|DeFT|31.27/0.8632|31.61/0.8801|31.34/0.8875| 30.31/0.8726|32.40/0.9247|
> >|LDT-S|31.2982/0.8600|31.4621/0.8793| 31.3679/0.8878|30.3735/0.8729| 32.4920/0.9228|
> >|LDT|31.36/0.8611|32.15/0.8880|31.51/0.8865|30.57/0.8705| 32.80/0.9246|
>
> **2.Reported gains over DeFT [1] are modest (e.g., Table 4), suggesting limited improvement relative to prior dual-branch approaches.**
>
> (1) In super-resolution tasks, the PSNR metric measures pixel-level differences, where performance improvements typically manifest as numerically modest gains. Therefore, the performance improvement of LDT over DeFT is actually significant. As shown in Table 5 of the manuscript (which corresponds to Table 4 in the previous version), LDT demonstrates consistent PSNR improvements over DeFT across all data branches, achieving a notable gain of 0.54 dB on the Sony data branch.
>
> (2) Furthermore, we compared LDT against DeFT across diverse task scenarios (super-resolution, classification, and semantic segmentation), various network architectures (ResNet18, ResNet50, ViT, Vision Mamba, MambaIR, HAT, SAFMN), and different data distributions (DRealSR, B100, Manga109, Set5, Set14, Urban100). LDT consistently achieves stable performance improvements in all these settings.

---

> ### Author Response · Authors · 2025-11-20
> **Official Comment by Authors [2/2]**
>
> **3.Sections 2.2.3 and 2.4 are unnecessarily long relative to their content, which impairs readability. Greater emphasis should be placed on more baseline comparisons against other domain generalization methods.**
>
> (1) In Section 2.2.3, our intention is to highlight the key distinction between the proposed LDT method and the existing DeFT approach—namely, our introduced layer-wise stable/unstable partitioning strategy. Additionally, to properly introduce the dynamic weight coefficient W used in our proposed DPU method, we first explain the fixed weight coefficient adopted by existing methods. Abbreviating this explanation may risk causing confusion to readers regarding the sudden appearance of the weight coefficient W and could lead to oversight of the core concept of layer-wise partitioning proposed in our work.
>
> (2) In Section 2.4, we first present the motivation behind the DPU method, followed by a detailed explanation of how the dynamic weight coefficient is determined.
>
>
> **4.Can you extend the evaluation to larger-scale and non-vision settings (e.g., ImageNet classification and an NLP benchmark such as WikiText-103), and outline any adaptations needed to apply LDT in these regimes?**
>
> (1) Validating domain generalization methods fundamentally requires at least two datasets with different distributions: one for training and another, with differing data distribution, for generalization testing (for classification tasks specifically, while distributions differ - such as daytime vs. nighttime - the output space must remain consistent, e.g., maintaining the same 1000 classes). Therefore, using ImageNet or WikiText-103 alone does not fulfill the necessary conditions to properly evaluate LDT's effectiveness in domain generalization.
>
> (2) As shown in Table 3 of this rebuttal, to verify LDT's effectiveness in NLP tasks, we utilized the Amazon review dataset specifically designed for NLP domain generalization research. It contains four data branches: DVD, Kitchen, Electronics, and Books. We used the first three as source domains and the last as the target domain. LDT demonstrates clear performance improvements compared to the DeFT method.
>
> (3) For classification tasks, we employed the VLCS dataset, specifically designed for domain generalization research in classification. It contains four data subsets: VOC, LabelMe, Caltech, and SUN. (see Appendix Table 14 in the revised manuscript).
>
> (4) For super-resolution tasks, we evaluated LDT on the DRealSR, B100, Manga109, Set5, Set14, and Urban100 datasets. (see Table 5 and Appendix Table 12 in the revised manuscript).
>
> (5) For semantic segmentation tasks, we verified LDT's effectiveness on the Cityscapes, BDD100K, and Mapillary datasets (see Appendix Table 15 in the revised manuscript).
>
> >Table 3 Domain Generalization on NLP Tasks
> >| Method|Source domain|Target domain: Book|
> >|-|-|-|
> >|FT||0.8725|
> >|DeFT|DVD, Kitchen, Electronics| 0.8800|
> >|LDT|| 0.9000|
>
>
> **5.Can you extend the evaluation to larger-scale and non-vision settings (e.g., ImageNet classification and an NLP benchmark such as WikiText-103), and outline any adaptations needed to apply LDT in these regimes?**
>
> (1) For image super-resolution tasks, we compared LDT with DeFT on the DRealSR dataset as well as the newly added B100, Manga109, Set5, Set14, and Urban100 datasets.
>
> (2) For NLP tasks, we compared LDT with DeFT on the Amazon review dataset.
>
> (3) For image classification tasks, we compared LDT with DeFT on the VOC, LabelMe, Caltech, and SUN datasets.
>
> (4) For semantic segmentation tasks, we compared LDT with DeFT on the Cityscapes, BDD100K, and Mapillary datasets.
>
> (5) Beyond comparing LDT and DeFT across different tasks, we also conducted comparisons between the two methods using various network architectures, including ResNet18, ResNet50, ViT, Vision Mamba, HAT, SAFMN, and MambaIR.

---

> ### Comment · Reviewer_GLbA · 2025-11-28
> **Response to the authors’ rebuttal**
>
> I thank the authors for their point-by-point and detailed rebuttal. I especially appreciate the extension of the theoretical analysis to the multi-layer setting added in the rebuttal, which helps better justify the method from a theoretical perspective.
>
> On the empirical side, however, I still feel that the absence of at least one large-scale benchmark weakens the strength of the evaluation. While I understand the authors's argument that WikiText or ImageNet baselines may not directly reflect LDT’s effectiveness in domain generalization and that the scope of the work is focused on generalization, I still believe the proposed method is fully applicable to such large-scale setups. Demonstrating improved performance on at least one large-scale baseline would significantly strengthen the case for the practical applicability of the model.
>
> Regarding the writing, the way notation is introduced in Sections 2.2.3 and 2.4 makes the presentation harder to follow. After each equation or block of equations, the authors devote an entire paragraph solely to listing and explaining the symbols and variables used in Equations (3), (4–6), (7–8), (9), (10), and (11). This could be made more concise by collecting the notation and definitions into a dedicated preliminaries/notation subsection before presenting the main equations, which would improve both readability and rigor.
>
> Similarly, for the theoretical results (Theorem 1) and the “Extension to Multi-Layer Network” section, I recommend that the authors first clearly state each theorem, then present the proof, and only afterward provide remarks or intuition. This would make the theoretical development feel more rigorous and structured, rather than like a continuous stream of explanation.
>
> After reading the feedback from the other reviewers, I acknowledge that my original evaluation was overly harsh. Nonetheless, given my remaining concerns about the current presentation and the scope of the experimental analysis, I am still unable to recommend an acceptance of the paper in its present form. That said, I appreciate the authors’ efforts to strengthen the theoretical results and to add further baselines during the rebuttal phase. I am therefore raising my overall score to 4 and have adjusted my evaluation of the other aspects of the work accordingly.
>
> Update: Due to a technical issue on OpenReview, I am currently unable to adjust my scores yet. I will revise the scores once the option becomes available.

---

> ### Author Response · Authors · 2025-11-29
> **Further Clarification**
>
> We appreciate your detailed feedback and the improved rating. We will address your new questions in a point-by-point manner below.
>
> **1. Clarification Regarding Large-Scale Benchmark Evaluation**
>
> (1) We have conducted experimental validation on large-scale datasets. The DRealSR dataset contains 61,274 LR-HR image pairs, which constitutes a substantial volume for super-resolution tasks. For semantic segmentation, the combined Cityscapes, BDD100K, and Mapillary datasets contain approximately 130,000 high-resolution images, representing a significant sample size for this task.
>
> (2) Since our primary research focus is domain generalization, we selected established benchmarks specifically designed for domain generalization tasks. This approach aligns with standard practice across published works in domain generalization [1-5].
>
> (3) Existing publications in domain generalization typically validate proposed methods on benchmarks from a single task scenario (either classification or segmentation). In contrast, we have demonstrated the effectiveness of our method across multiple tasks: image classification, super-resolution, semantic segmentation, and NLP tasks. This comprehensive multi-task evaluation provides robust evidence for our method's generalization capability.
>
> (4) Current ultra-large-scale datasets designed for foundation models generally lack distribution-based partitioning, as their test and training distributions remain relatively similar, making them unsuitable for properly evaluating domain generalization methods. Furthermore, training networks on ultra-large-scale datasets requires substantial time investment, which presents significant challenges for completion within the limited timeframe of the author-reviewer discussion period. Additionally, our research is not specifically targeted at large foundation models.
>
> **2. Regarding the writing, the way notation is introduced in Sections 2.2.3 and 2.4 makes the presentation harder to follow. After each equation or block of equations, the authors devote an entire paragraph solely to listing and explaining the symbols and variables used in Equations (3), (4–6), (7–8), (9), (10), and (11). This could be made more concise by collecting the notation and definitions into a dedicated preliminaries/notation subsection before presenting the main equations, which would improve both readability and rigor.**
>
> We have revised the manuscript by adding a new subsection titled "2.1 Problem Definition and Notation" that provides a unified explanation of all mathematical symbols used throughout the paper, thereby significantly improving readability.
>
> **3. Similarly, for the theoretical results (Theorem 1) and the “Extension to Multi-Layer Network” section, I recommend that the authors first clearly state each theorem, then present the proof, and only afterward provide remarks or intuition. This would make the theoretical development feel more rigorous and structured, rather than like a continuous stream of explanation.**
>
> We have revised both Theorem 1 and Theorem 2 in the "Extension to Multi-Layer Network" section by clearly defining each theorem and its corresponding proof, further enhancing the readability of the paper.
>
>
> **References**
>
> [1] Decoupled Finetuning for Domain Generalizable Semantic Segmentation
>
> [2] Style blind domain generalized semantic segmentation via covariance alignment and semantic consistence contrastive learning
>
> [3] Fine-tuning can distort pretrained features and underperform out-of-distribution
>
> [4] DGMamba: Domain Generalization via Generalized State Space Model.
>
> [5] Start: A generalized state space model with saliency-driven token-aware transformation

---

### Official Review · Reviewer_xH6P · 2025-11-02

**Soundness:** 3
**Presentation:** 2
**Contribution:** 3
**Rating:** 6
**Confidence:** 4

**Summary:**

This paper attempts to mitigate the influence of unstable parameters on stable parameters for domain generalization. The authors proposed a fine-grained layer-wise partitioning strategy to improve parameter update stability, named Layer-Decomposition Training (LDT). LDT separates and cross-froze the stable and unstable layers during training, thereby mitigating the perturbations from unstable layers. Besides, the authors propose Dynamic Parameter Update (DPU) strategy to adjust the update coefficients of diverse layers according to their fluctuation amplitudes. Experiments on super-resolution, classification, and semantic segmentation demonstrate that LDT could enhance the generalization performance.

**Strengths:**

[+] The paper is easy to follow. \
[+] The detail of the method and experiment is well described. \
[+] The related work is detailed.

**Weaknesses:**

Major weakness:

[-] The explanations for the experimental results are insufficient. This is particularly evident in Figure 2, Table 4, and Appendix Figure 5.

[-] The authors claim that "the same parameter update coefficient would inevitably result in information loss and counterintuitive behavior." Please clarify what is meant by "information loss" and provide evidence to verify this phenomenon.

[-] The number of benchmarks used across different tasks is limited. Expanding the benchmark set would help better validate the effectiveness of the proposed approach.

[-] Regarding Table 2:
(1) The performance differences among the methods—particularly between Mean, Var/Mean, and Var—are not significant. Please explain the possible reasons for this.
(2) Why does the Var/Mean method, which is supposed to better reflect parameter fluctuations without magnitude interference, underperform compared to Var?

[-] In Appendix Table 7, the average performance of ResNet50 + DeFT (0.7978) is lower than that of ResNet18 + DeFT (0.7992). Please explain this counterintuitive result.

[-] The authors selected layer partitioning ratios of 0.4 or 0.5 for unstable layers. Is this ratio also optimal for other tasks, such as classification and semantic segmentation?

Minor weakness:

[-] Typographical errors should be corrected to ensure clarity. For instance, "0.8746 pm 4.79..." in Table 1 should be revised to "0.8746 ± 4.79...".

**Questions:**

Please refer to the weaknesses.

---

> ### Author Response · Authors · 2025-11-20
> **Official Comment by Authors [1/2]**
>
> We thank you for your detailed feedback. We will address all of your comments and questions point by point below.
>
> **1.The explanations for the experimental results are insufficient. This is particularly evident in Figure 2, Table 4, and Appendix Figure 5.**
>
> We have revised the manuscript to provide more comprehensive explanations for the experimental results, particularly concerning Figure 2, Table 4, and Appendix Figure 5. The specific amendments are as follows:
>
> (1) Revisions in Section 1 (Problem Discovery):
>
> * Insufficient granularity in partitioning stable and unstable parameters: existing methods such as LP-FT and DeFT designate the backbone network as the stable layers and the prediction head as the unstable layers. However, as shown in Figure 2a, certain layers within the backbone network demonstrate higher gradient variance compared to the prediction head, indicating that some backbone layers are actually less stable than the head. These unstable parameters in the backbone substantially influence parameter updates across all other layers. As a result, existing partitioning methods based on the backbone-head distinction operate at an insufficient granularity, leading to misclassification issues.
> * Inadequate adaptability in parameter updates: even within the unstable layers, significant disparities in gradient variance exist (with the ratio between the minimum and maximum standard deviations approaching 10×, as shown in Figure 2b). Existing methods apply the same update strategies to all layers, despite their varying gradient standard deviations, which inevitably leads to underutilization of information. This issue is even more pronounced among stable layers, as illustrated in Figure 2c.
>
> (2) Revisions in Section 3.4 (Comparative Experiments):
>
> As shown in Table.5 (Table 4 in the previous version), we compare our proposed LDT method with other domain generalization methods, including parameter-correlation-focused methods LP-FT and DeFT, feature-perturbation-based domain generalization methods START, DTAM, and Wang et al., as well as domain adaptation methods IODA and SRTTA trained on both source and target domains. For instance, compared to DeFT, LDT obtains a PSNR improvement of 0.54 dB on the Sony data branch. Visual comparisons in Appendix Figure 5 further confirm that the super-resolved images produced by LDT exhibit clearer and more accurate texture details.
>
> **2.The authors claim that "the same parameter update coefficient would inevitably result in information loss and counterintuitive behavior." Please clarify what is meant by "information loss" and provide evidence to verify this phenomenon.**
>
> We thank the reviewer for raising this point. The term "information loss" used in our original statement specifically refers to the underutilization of information. This occurs because significant gradient variance disparities exist within unstable layers, yet existing methods apply the same update strategies across all layers, regardless of their differing gradient standard deviations. As a result, the valuable gradient standard deviation information from each layer is not fully exploited. We will revise the description to "underutilization of information" in the revised manuscript to ensure clarity and precision.
>
> **3.The number of benchmarks used across different tasks is limited. Expanding the benchmark set would help better validate the effectiveness of the proposed approach.**
>
> (1) As shown in Tables 1 and 2 of this rebuttal, we have supplemented the evaluation of LDT with additional benchmarks for image super-resolution and NLP tasks respectively.
>
> (2) To date, we have validated the effectiveness of our approach across the following diverse task scenarios and data distributions:
> * For image super-resolution, the effectiveness of LDT has been evaluated on the DRealSR, B100, Manga109, Set5, Set14, and Urban100 datasets.
> * For image classification, LDT has been validated on the VOC, LabelMe, Caltech, and SUN datasets.
> * For semantic segmentation, the effectiveness of LDT has been verified on the Cityscapes, Mapillary and BDD100K datasets.
> * For NLP tasks, LDT has been validated on the Amazon review dataset.
>
> (3) Given the limited duration of the author-reviewer discussion window and the need to address comments from multiple reviewers, we would be pleased to incorporate any additional benchmark sets the reviewer may suggest in an extended version of this work.
>
> >Table 1. Performance Comparison on Different Data Distributions for Image Super-Resolution
> >|Method|B100|Manga109|Set5|Set14|Urban100|
> >|-|-|-|-|-|-|
> >|LPFT|24.13/0.6551|23.79/0.8135|25.19/0.7713|23.58/0.6735|21.77/0.6717|
> >|DeFT|25.14/0.6885|24.53/0.8306|26.29/0.8001|24.94/0.7157|22.91/0.7113|
> >|LDT|25.38/0.6938|24.70/0.8467|26.44/0.8100|25.03/0.7214|23.03/0.7261|
>
> >Table 2. Domain Generalization on NLP Tasks
> >|Method|Source domain|Target domain: Book|
> >|-|-|-|
> >|FT||0.8725|
> >|DeFT|DVD, Kitchen, Electronics|0.8800|
> >|LDT||0.9000|

---

> ### Author Response · Authors · 2025-11-20
> **Official Comment by Authors [2/2]**
>
> **4.Regarding Table 2: (1) The performance differences among the methods—particularly between Mean, Var/Mean, and Var—are not significant. Please explain the possible reasons for this. (2) Why does the Var/Mean method, which is supposed to better reflect parameter fluctuations without magnitude interference, underperform compared to Var?**
>
> (1) (a)In super-resolution tasks, the PSNR metric measures pixel-level differences. Therefore, even numerically small performance variations can correspond to perceptibly significant differences in output quality. (b) Unstable parameters are highly sensitive to data distribution shifts. Slight changes in data distribution can cause substantial fluctuations in these parameters, leading to large gradient means. As a result, using the mean alone can partially identify unstable layers. However, parameters with high learning efficiency may also produce large gradients when encountering different data distributions to achieve rapid fitting. A partitioning strategy based solely on mean values would misclassify such layers as unstable and freeze them, thereby reducing fitting efficiency. As shown in Table 3 of the main manuscript (which corresponds to Table 2 in the previous version), the mean-based partitioning method underperforms compared to the variance-based approach.
>
> (2) As mentioned above, due to the sensitivity of unstable parameters, their gradients often exhibit large mean values. This results in artificially low Var/Mean ratios, causing these layers to be misclassified as stable. Such misclassification ultimately reduces the network's generalization performance.
>
> **5.In Appendix Table 7, the average performance of ResNet50 + DeFT (0.7978) is lower than that of ResNet18 + DeFT (0.7992). Please explain this counterintuitive result.**
>
> Taking datasets A and B as an example, in domain generalization tasks, models are trained on dataset A and tested on dataset B. Since ResNet50 has stronger feature learning capability than ResNet18, it tends to achieve a higher degree of fitting to dataset A during training. However, when the distribution of dataset B differs from that of dataset A, the model that performs better on dataset A (ResNet50) may exhibit greater performance degradation on dataset B due to overfitting to the training dataset A. This can lead to situations where a more powerful architecture shows lower generalization performance on unseen target domains.
>
> **6.The authors selected layer partitioning ratios of 0.4 or 0.5 for unstable layers. Is this ratio also optimal for other tasks, such as classification and semantic segmentation?**
>
> (1) As shown in Table 3 of this rebuttal, we have supplemented experiments that analyze the impact of different partitioning ratios on model performance in semantic segmentation tasks.
>
> (2) Regarding the selection method for the stable/unstable layer partitioning ratio, we first execute the first stage of the LDT method: we compute and rank the gradient variances of each network layer across 300 samples, then visualize them to identify the variance jump point (which requires meeting two conditions: rapid variance increase and location near the middle of the overall ranking). In SR tasks using Mamba networks, the variance jump point is near 0.4, while in semantic segmentation tasks using ResNet networks, the variance jump point is near 0.7.
>
>
> >Table 3. Ablation Study on the Impact of Layer Partitioning Ratios for Semantic Segmentation. This study uses Cityscapes as the source domain, with BDD100K and Mapillary serving as Target Domain 1 and Target Domain 2, respectively.
> >|Ratio| Target domain 1 (mIOU) | Target domain 2 (mIOU) |
> >|-|-|-|
> >|0.1|38.155|43.852|
> >|0.2|38.452|42.797|
> >|0.3|42.587|50.547|
> >|0.4|44.851|53.075|
> >|0.5|44.766|53.647|
> >|0.6|43.284|53.687|
> >|0.7|45.420|55.733|
> >|0.8|44.445|54.448|
> >|0.9|45.386|55.444|
>
>
> **7.Typographical errors should be corrected to ensure clarity. For instance, "0.8746 pm 4.79..." in Table 1 should be revised to "0.8746 ± 4.79...".**
>
> We have corrected this error in the manuscript.

---

### Comment · Area_Chair_XxC1 · 2025-11-27
**Reminder: Engage in Discussions and Finalize Your Rating**

Dear Reviewers,

Thank you for your valuable reviews. With the Reviewer-Author Discussions deadline approaching, please take a moment to read the authors’ rebuttal and the other reviewers’ feedback, and participate in the discussions and respond to the authors. Finally, be sure to complete the “Final Justification” text box and update your “Rating” as needed. Your contribution is greatly appreciated. I will flag irresponsible (final) reviews and/or any reviewers not participating in discussions.

Reviewers are expected to stay engaged in discussions, initiate them, respond to authors’ rebuttal, ask questions, and listen to answers to help clarify remaining issues.

It is not OK to stay quiet.

It is not OK to leave discussions till the last moment.

If authors have resolved your (rebuttal) questions, do tell them so.

If authors have not resolved your (rebuttal) questions, do tell them so too.

Thanks,

AC

---

### Author Response · Authors · 2025-12-01
**Summary of Review Comments and Rebuttal [1/2]**

Our work has been positively acknowledged by all reviewers:

* Reviewer xH6P noted: "The paper is easy to follow..."

* Reviewer GLbA commented: "...is easy to follow and easy to implement in practice."

* Reviewer e1yc highlighted: "The idea...is both conceptually novel and practically meaningful...Theoretical motivation is well supported."

* Reviewer cWBr emphasized: "Intuitive Problem Definition...Excellent Versatility (Task- & Architecture-Agnostic)...Robust Experimental Validation."

* Reviewer kBcr acknowledged: "Clear problem framing...Method simplicity & generality..."


**Initial Ratings and Changes**

Initial ratings: xH6P: 6, GLbA: 2, e1yc: 4, cWBr: 6, kBcr: 6

Revised ratings: xH6P: 6, GLbA: 4, e1yc: 4, cWBr: 6, kBcr: 6

1. **Reviewer GLbA acknowledged that we have addressed their initial concerns and raised their rating to 4. They also pointed out several new concerns, indicating that further rating improvements could be achieved if these concerns are resolved.**
Due to an OpenReview API bug, the reviewer was unable to respond to our latest clarifications. However, we believe our updated responses effectively address the newly raised questions.

2. Before the OpenReview API bug occurred, Reviewer kBcr confirmed that we had addressed all of their concerns.

3. Due to the constraints of the OpenReview API bug, the remaining reviewers were unable to provide a timely response to our rebuttal.

Note: Due to the addition of Table 1, the table numbers mentioned in the reviewer's comments now correspond to the table numbers in the revised manuscript with an offset of +1. For instance, the reviewer's reference to "Table 2" corresponds to "Table 3" in the revised manuscript.

---

> ### Author Response · Authors · 2025-12-01
> **Summary of Review Comments and Rebuttal [2/2]**
>
> Below, we summarize the reviewers' comments and our corresponding rebuttals.
>
> # xH6P
> Initial rating:6
>
> >|Concern|Rebuttal|
> >|-|-|
> >|Interpretation of experimental results|We have added corresponding explanations to the main manuscript.|
> >|Misinterpretation of the statement: "the same parameter update coefficient would inevitably result in information loss and counterintuitive behavior."|We have clarified the intended meaning and revised the description in the main manuscript.|
> >|Request for additional benchmarks|We have added the following benchmarks: (1) For Super-Resolution tasks: B100, Manga109, Set5, Set14, and Urban100. (2) For NLP tasks: the Amazon review dataset.|
> >|Interpretation of ablation studies:(1) Var/Mean vs. Var methods. (2) ResNet50 vs. ResNet18|We have provided detailed explanations of both ablation study results in this rebuttal.|
> >|Determination of optimal layer partitioning ratios across different tasks|We have conducted additional ablation studies on layer partitioning ratios for semantic segmentation, completing comprehensive experiments for both image super-resolution and semantic segmentation tasks.|
> >|Typographical errors (e.g., 0.8746 pm 4.79...)|We have corrected this error in the main manuscript.|
>
> # GLbA
> Initial rating:2
>
> **Reviewer GLbA acknowledged that we have addressed their initial concerns and raised their rating to 4. They also pointed out several new concerns, indicating that further rating improvements could be achieved if these concerns are resolved.**
> Due to an OpenReview API bug, the reviewer was unable to respond to our latest clarifications. However, we believe our updated responses effectively address the newly raised questions.
>
>
>
> >|Concern|Rebuttal|
> >|-|-|
> >|Validation on large-scale datasets|We have conducted experimental validation on large-scale datasets DRealSR, Cityscapes, BDD100K and Mapillary. The DRealSR dataset contains 61,274 LR-HR images, which constitutes a substantial volume for super-resolution tasks. For semantic segmentation, the combined Cityscapes, BDD100K, and Mapillary datasets contain approximately 130,000 high-resolution images, representing a significant sample size for this task.|
> >|Suggestion to add a dedicated preliminaries/notation subsection|We have revised the manuscript by adding a new subsection titled "2.1 Problem Definition and Notation," which provides a unified explanation of all mathematical symbols used throughout the paper, thereby significantly improving readability.|
> >|Recommendation to optimize proof descriptions with clear definitions of theorems and proofs|We have revised both Theorem 1 and Theorem 2 in the "Extension to Multi-Layer Network" section by clearly defining each theorem and its corresponding proof, further enhancing the readability of the paper.|
>
> # e1yc
> Initial rating:4
>
> >|Concern|Rebuttal|
> >|-|-|
> >|Suggestion to extend the "two-layer module proof" to "multi-layer module proof"|We have extended the theoretical derivation of gradient interference to the multi-layer case (Appendix B).|
> >|Analysis on scalability to larger networks (e.g., ViT-L or Swin-L) is missing|We have added scalability analysis for larger networks and conducted comparative experiments on computational efficiency with baseline models, existing methods, and variants of our approach.|
> >|Additional ablation study on normalization-based scaling instead of discrete ranking|We have supplemented the ablation study with normalization-based scaling experiments.|
> >|Suggestion to explore single-network architecture|We have further improved our method by proposing LDT-S, a single-network branch version of our method, which achieves even better computational efficiency.|
> >|Discussion on potential applications in the era of large models|We have provided a perspective on the potential applications of our method in the context of large models.|
>
> # cWBr
> Initial rating:6
>
> >|Concern|Rebuttal|
> >|-|-|
> >|Clarification on the hyperparameter "Ratio^{u}"|We have provided a detailed explanation of the methodology for selecting this hyperparameter.|
> >|Suggestion to add ablation study validating the use of variance values to determine weight coefficients|We have supplemented the requested ablation experiment.|
> >|Exploration of methods to reduce computational requirements|1. We have improved our proposed method by introducing LDT-S, a single-network branch variant that significantly reduces computational load. 2. We have also discussed several approaches mentioned by the reviewer, such as distillation.|
> >|Suggestion to modify the static layer partitioning strategy to a dynamic one|We have conducted ablation studies with a dynamic layer partitioning strategy and found that it effectively enhances network performance.|
>
> # kBcr
> Initial rating: 6
>
> **Before the OpenReview API bug occurred, Reviewer kBcr confirmed that we had addressed all of their concerns.** Therefore, we have not provided a detailed summary of the questions and responses related to this reviewer.

---

### Meta-Review · Area_Chair_3qCv · 2026-01-05

**Summary:**

The reviewers identified several strengths of the paper, including the clarity of presentation of the technical content, the ease of the implementation and the general applicability of the method. At the same time, they have identified several weaknesses, including novelty (citing the existing work of dual branch training), limited scale/breadth of experiments and theory, and potential sensitivity to hyper-parameters, in addition to other minor concerns.

**Reviewer Concerns:**

The authors addressed the novelty issue by clarifying the differences and also introducing a single-branch variant. They also extended the scope of their theoretical analysis. To address the limited scope of their numerical experiments, the authors added additional benchmarks in super-resolution and NLP tasks.

**Reviewer Scores:**

The core concerns of the reviewers have be sufficiently addressed, which improved the paper. I believe that the reviewers would have converged on a positive (albeit still borderline, in view of existing works) rating.

---

### Decision · Program_Chairs · 2026-01-26

Accept (Poster)